# Counterfactual Implicit Feedback Modeling

**Chuan Zhou**[1,2]    **Lina Yao**[3,4]    **Haoxuan Li**[5,2,*]    **Mingming Gong**[1,2,*]

[1]The University of Melbourne    [2]Mohamed bin Zayed University of Artificial Intelligence
[3]The University of New South Wales    [4]CSIRO's Data61    [5]Peking University
chuan.zhou@student.unimelb.edu.au, lina.yao@data61.csiro.au,
hxli@stu.pku.edu.cn, mingming.gong@unimelb.edu.au

## Abstract

In recommendation systems, implicit feedback data can be automatically recorded and is more common than explicit feedback data. However, implicit feedback poses two challenges for relevance prediction, namely (a) **positive-unlabeled** (PU): negative feedback does not necessarily imply low relevance and (b) **missing not at random** (MNAR): items that are popular or frequently recommended tend to receive more clicks than other items, even if the user does not have a significant interest in them. Existing methods either overlook the MNAR issue or fail to account for the inherent mechanism of the PU issue. As a result, they may lead to inaccurate relevance predictions or inflated biases and variances. In this paper, we formulate the implicit feedback problem as a counterfactual estimation problem with missing treatment variables. Prediction of the relevance in implicit feedback is equivalent to answering the counterfactual question that "*whether a user would click a specific item if exposed to it?*". To solve the counterfactual question, we propose the Counterfactual Implicit Feedback (Counter-IF) prediction approach that divides the user-item pairs into four disjoint groups, namely definitely positive (DP), highly exposed (HE), highly unexposed (HU), and unknown (UN) groups. Specifically, Counter-IF first performs missing treatment imputation with different confidence levels from raw implicit feedback, then estimates the counterfactual outcomes via causal representation learning that combines pointwise loss and pairwise loss based on the user-item pairs stratification. Theoretically the generalization bound of the learned model is derived. Extensive experiments are conducted on publicly available datasets to demonstrate the effectiveness of our approach. The code is available at `https://github.com/zhouchuanCN/NeurIPS25-Counter-IF`.

## 1  Introduction

Recommender systems are technical tools that analyze historical user behavioral data to predict their preferences and actively provide personalized recommendations [1, 2, 3], which have been widely applied to various fields, such as e-commerce, streaming video, and social media [4, 5, 6, 7]. There are two types of feedback mechanisms in user behavioral data, including explicit feedback [8, 9, 10], which refers to the explicit signals that users directly express their preferences, such as product ratings. Implicit feedback [11], on the other hand, comes from the indirect interactions between users and the system, including unstructured behavioral trajectories such as the length of time spent on the page, clicks, and so on. Implicit feedback has a great advantage over explicit feedback in terms of data acquisition [12, 5], as most user behaviors are implicit [13], which can be collected continuously on a large scale and can reflect the potential user demand more promptly.

In implicit feedback recommendation, the goal is to infer the user's true relevance or preference for an item from their behavior data, i.e., to determine whether the user is likely to be interested in an item.

---

*Haoxuan Li and Mingming Gong are the corresponding authors.

However, this process faces two key challenges. The first one is the **positive-unlabeld** (PU) learning problem [14, 15]. The system only observes a binary signal of whether the user clicks the item, but the clicking behavior does not fully reflect user preferences. Specifically, the items not clicked on may be attributed to user disinterest, or they may not even be exposed to the user. The second challenge is that the feedback data is **missing not at random** (MNAR) [16, 17]. For example, items frequently recommended tend to attract more clicks, even if the user does not have a substantial interest in them. In recent years, studies have begun to notice that MNAR in implicit feedback data can introduce bias into predictions, thus damaging the performance of the recommendation system [18, 19, 20].

In the development of implicit feedback recommendation, early approaches such as weighted matrix factorization (WMF) [21], exposure perception matrix factorization (ExpoMF) [8] and Bayesian personalized ranking (BPR) [13] are based on the key assumption that the observed feedbacks directly reflect the true preferences, whereas unobserved feedbacks are uniformly treated as negative samples or given low weights. However, early methods assume that the missing mechanism is missing completely at random (MCAR), ignoring the MNAR nature of implicit data, leading to biased relevance estimation [16]. Recent studies attempt to introduce causal techniques into implicit feedback modeling, e.g., through counterfactual bias correction using propensity score [22, 23]. However, these causal learning methods still have some drawbacks. The first one is that propensity models are prone to be overly confident, generating extremely inaccurate propensity score estimation [24, 25, 26]. And like any propensity-based method, the bias and variance of the estimator can be extremely large with small propensity [27, 28], thus affecting the effectiveness of relevance prediction. The second one is that existing causal recommendation methods for implicit feedback data usually use EM algorithms to estimate the exposure propensity, but the intrinsic positive-unlabeled mechanism of implicit feedback is neglected, which may hinder the model from accurately estimating the propensity.

In this paper we propose that implicit feedback recommendation can be addressed by answering the counterfactual question: **whether a user would click a specific item if exposed to it?** To tackle this problem, we formulate it as a counterfactual outcome prediction problem with missing treatment. Let the feedback label be the outcome, and the exposure be the treatment. To answer the counterfactual question, we actually need to estimate the potential outcome corresponding to the exposed treatment group, with treatment of only the exposed group observed. Although there has been work dealing with the MNAR problem with explicit feedback data [27, 29, 30, 31], and remarkably much more work in the area of statistics estimating causal effects [32, 33], none of them could be applied to the counterfactual problem formulated in this paper. That is because all previous causal methods require the treatment to be observed for every sample in the dataset. However, this counterfactual problem brings unique challenges with missing treatment.

To overcome these challenges, we propose the **Counter**factual **I**mplicit **F**eedback (Counter-IF) method for estimating counterfactual outcomes in the presence of missing treatment variables in implicit feedback scenarios. To uncover more information from negative samples based on the positive-unlabeled nature, the Counter-IF consists of stratifying user-item pairs in implicit feedback. Specifically, using the estimated confidence, we impute the exposure only for samples with high confidence. Apart from the **definitely positive** (DP) samples group, we classify the negative samples into three groups based on the different reasons leading to unclick: those with **high probability of being exposed** (HE) tend to have low relevance and those with **high probability of being unexposed** (HU) tend to have higher relevance than other **unknown** (UN) negative samples. Counter-IF also includes a causal representation learning framework that combines pointwise and pairwise losses based on the imputed treatments, leading to accurate counterfactual outcomes estimation.

The main contributions can be summarized as follows:

- We are the first to formalize the relevance prediction problem under implicit feedback scenarios as a counterfactual outcome estimation problem with missing treatments.

- We propose a sample stratification algorithm in Counter-IF for implicit feedback using a treatment variable imputation method with confidence, reflecting different mechanisms of negative sample generation.

- We propose a causal representation learning framework in Counter-IF to answer the formalized counterfactual questions. We theoretically derive the generalization bound of our causal learning model.

- We conduct extensive experiments on publicly available real-world datasets, demonstrating the proposed Counter-IF significantly outperforms state-of-the-art methods.

## 2    Problem Setup of Implicit Feedback

Let $u \in \mathcal{U}$ denote a user, $i \in \mathcal{I}$ be an item and $\mathcal{D} = \mathcal{U} \times \mathcal{I}$ be the set of all user-item pairs. The complete set consists of $|\mathcal{U}| \times |\mathcal{I}|$ user-item pairs. The recommender system observes only the implicit feedback $Y_{u,i} \in \{0,1\}$, which passively captures whether the user $u$ clicks the item $i$. The feature of $(u,i)$ is denoted as $X_{u,i}$. We divide all user-item pairs according to $Y_{u,i}$, i.e., $\mathcal{D}_1 = \{(u,i) \mid (u,i) \in \mathcal{D}, Y_{u,i} = 1\}$ and $\mathcal{D}_0 = \{(u,i) \mid (u,i) \in \mathcal{D}, Y_{u,i} = 0\}$. Due to the sparsity of positive samples in implicit feedback data, we have $|\mathcal{D}_1| \ll |\mathcal{D}_0|$. A positive feedback $Y_{u,i} = 1$ directly implies that $u$ and $i$ are relevant. However, we are not certain whether a negative feedback $Y_{u,i} = 0$ indicates the item is irrelevant to the user since it depends on whether $u$ is exposed to $i$.

We consider the exposure $O_{u,i}$ as a treatment, with $O_{u,i} = 1$ meaning that the user $u$ is exposed to the item $i$ and vice versa. Note that we cannot observe $O_{u,i}$ for all the user-item pairs in $\mathcal{D}$. Instead, we can merely infer that for those $(u,i) \in \mathcal{D}_1$, the user $u$ must be exposed to item $i$, thus $O_{u,i} = 1$. However, for those $(u,i) \in \mathcal{D}_0$, $O_{u,i}$ is unknown. To more clearly formulate the implicit feedback problem, we introduce the true relevance score $S_{u,i} \in \{0,1\}$, with $S_{u,i} = 1$ indicating that $u$ and $i$ are relevant and vice versa. Following previous work [16], we assume in this paper that:

$$Y_{u,i} = O_{u,i} \cdot S_{u,i}. \tag{1}$$

Given $Y_{u,i} = 1, O_{u,i} = 1$ for $(u,i) \in \mathcal{D}_1$ and $Y_{u,i} = 0$ for $(u,i) \in \mathcal{D}_0$, we would like to predict $S_{u,i}$ for all $(u,i) \in \mathcal{D}$. Note that Equation (1) implies $Y_{u,i} = 1 \iff O_{u,i} = 1$ and $S_{u,i} = 1$.

## 3    Methodology

### 3.1    Causal Formulation of Implicit Feedback

The question we aim to answer in implicit feedback recommendation is "whether a user would click a specific item if exposed to it", a typical counterfactual question. Therefore, it is logical to express it in causal terminology. In this paper, we employ the potential outcome framework. In our causal formulation, we let $O_{u,i}$ be the treatment, and $Y_{u,i}(1)$ be the potential outcome of feedback if forcing $O_{u,i} = 1$, and $Y_{u,i}(0)$ be the potential outcome of feedback if we force $O_{u,i} = 0$. We require the consistency assumption, which means that if $u$ is exposed to $i$, the observed outcome of feedback is the potential outcome we aim to estimate, i.e., $Y_{u,i} = O_{u,i}Y_{u,i}(1) + (1 - O_{u,i})Y_{u,i}(0)$. We

Table 1: The user-item pairs are divided into four subgroups from a principal stratification perspective, named *Definitely Positive* group, *Highly Exposed* group, *Highly Unexposed* group, and *Unknown* group. The red font indicates that the value is imputed.

| Group | $O$ | $R$ | $Y$ | $Y(1)$ |
|---|---|---|---|---|
| Definitely Positive (DP) | 1 | 1 | 1 | 1 |
| Highly Exposed (HE) | 1 | 1 | 0 | 0 |
| Highly Unexposed (HU) | 0 | 1 | 0 | ? |
| Unknown (UN) | ? | 0 | 0 | ? |

also assume that the stable unit treatment value assumption (SUTVA) holds, i.e., there should be only one form of exposure between $u$ and $i$, and there is no interference between user-item pairs. In addition, we assume that the unconfoundedness assumption holds, i.e., $Y_{u,i}(1) \perp\!\!\!\perp O_{u,i}|X_{u,i}$. In the recommendation scenario, this assumption implies that the intrinsic relevance between $u$ and $i$ is independent of whether to expose $u$ to $i$. Then on the one hand, by the consistency assumption we have $O_{u,i} = 0 \Rightarrow Y_{u,i} = Y_{u,i}(0)$. Meanwhile, by Equation (1), we have $O_{u,i} = 0 \Rightarrow Y_{u,i} = 0$. As a consequence, $Y_{u,i}(0) = 0$ for $(u,i) \in \mathcal{D}$. On the other hand, we have $O_{u,i} = 1 \Rightarrow Y_{u,i} = Y_{u,i}(1)$ and $O_{u,i} = 1 \Rightarrow Y_{u,i} = S_{u,i}$. Combining the two equations derived under the condition $O_{u,i} = 1$, we have $Y_{u,i}(1) = S_{u,i}$ for $(u,i) \in \mathcal{D}$.

Therefore, we transform the estimation of $S_{u,i}$ into the estimation of $Y_{u,i}(1)$. In other words, we formulate the implicit feedback problem as **missing treatment counterfactual problem**, where the treatment assignments to the samples with negative feedback are unknown, and we aim to estimate the potential outcomes $Y_{u,i}(1)$. With the consistency assumption, we can infer that the potential outcome $Y_{u,i}(1) = 1$ for $(u,i) \in \mathcal{D}_1$, since for these samples $Y_{u,i} = 1$, we can infer $O_{u,i} = 1$, then $Y_{u,i}(1) = Y_{u,i} = 1$. However, for $(u,i) \in \mathcal{D}_0$, we are not sure whether $O_{u,i} = 1$ or $O_{u,i} = 0$, so we cannot directly infer the value of $Y_{u,i}(1)$ for this group of user-item pairs.

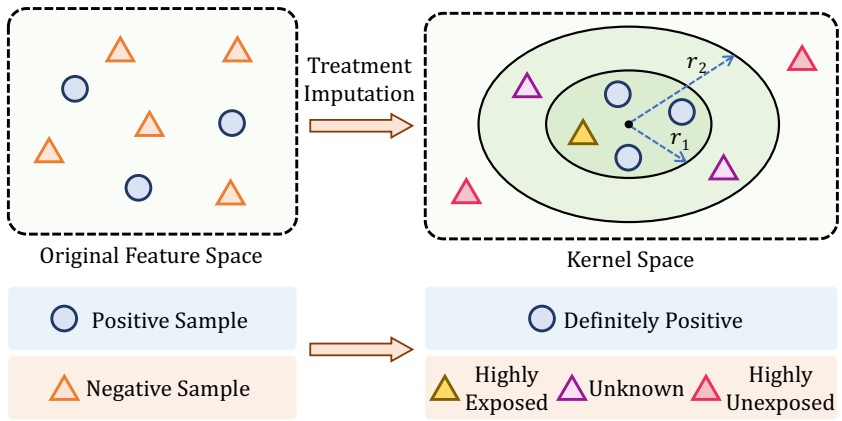

Figure 1: Missing treatment imputation with different confidence level from implicit feedback.

## 3.2 Stratification of User-Item Pairs

For the purpose of counterfactual estimation, it is almost impossible to solely rely on the outcome variables and a small portion of the treatment variables. So we, therefore, wanted to dig deeper into the intrinsic mechanisms of the implicit feedback data to get more information about exposure. Unlike previous methods that impute treatments using a consistent scheme for all samples in $\mathcal{D}$, based on the diversity of meanings of negative samples in implicit feedback, we consider imputation only for those items the system is highly confident to recommend to a user. Specifically, we define a binary indicator $R_{u,i} \in \{0, 1\}$, which reflects the confidence of the recommender system in assigning the treatment. $R_{u,i} = 1$ means that the system is with high confidence, and $R_{u,i} = 0$ means that the system is not confident about treatment assignment. We denote the estimated confidence as $\hat{R}_{u,i}$ for our treatment imputation model. For those with high confidence, we impute the treatment as the predicted $\hat{O}_{u,i}$. Then based on $(\hat{O}_{u,i}, \hat{R}_{u,i}, Y_{u,i})$, we divide all user-item pairs into four strata:

- Definitely Positive (DP) Group $\mathcal{D}_{DP}$: This group consists of user-item pairs with $Y_{u,i}(1) = 1$, the imputation method should be definitely confident about the corresponding treatment $O_{u,i} = 1$, meaning $u$ has definitely been exposed to $i$. We have $\mathcal{D}_{DP} = \mathcal{D}_1$.

- For user-item pairs in $\mathcal{D}_0$, we further classify them into three categories, according to the estimated confidence indicator $\hat{R}_{u,i}$ and imputed treatment $\hat{O}_{u,i}$:

  - Highly Exposed (HE) Group: The imputation method has high confidence that the user is exposed to the item ($\hat{R}_{u,i} = 1$ and $\hat{O}_{u,i} = 1$), thus impute the treatment as 1, meaning the user tend to be exposed to the item ($O_{u,i} = 1$). However, the observed $Y_{u,i} = 0$, which means that $Y_{u,i}(1)$ is likely to be 0.
  - Highly Unexposed (HU) Group: The imputation method has high confidence that the user is unexposed to the item ($R_{u,i} = 1$ and $\hat{O}_{u,i} = 0$), thus impute the treatment as 0. $Y_{u,i}(1)$ is more likely to be 1 than other negative samples.
  - Unknown (UN) Group: The imputation method has low confidence to assign treatment ($R_{u,i} = 0$), so we do not impute for the sample ($O_{u,i} = ?$), keeping its multiple interpretation nature.

As Table 1 shows, we can infer the value of $Y_{u,i}(1)$ for the DP and HE groups, but not for the HU and UN groups. For the DP group $Y_{u,i}(1) = Y_{u,i} = 1$. For the HE group, given the imputation is accurate, we have $Y_{u,i}(1) = Y_{u,i} = 0$.

## 3.3 Treatment Imputation with Confidence

Intuitively, we expect the DP group, consisting of user-item pairs known to be exposed, and the HE group, which includes user-item pairs for whom the system has high confidence that they are exposed, to be close to each other in a specific feature space. On the other hand, the HU group, composed of user-item pairs for whom the system is confident that are unexposed, should be far

away from the exposed groups (HE & DP). The remaining users, whose exposure status remains uncertain, form the Unknown group (UN), positioned between the positive and negative groups. To divide users into the four subgroups, we propose a novel exposure estimation approach to impute the treatments for solving the counterfactual problem, as shown in Figure 1. Inspired by support vector domain description [34], we propose a novel treatment imputation method that utilizes the distance requirement between the four groups to assign samples to groups. Specifically, with feature $x_{u,i} \in \mathcal{X}$ as the input, we enclose a proportion of $\alpha$ samples of DP within a hypersphere, where $\alpha \in (0, 1)$ is a hyperparameter. Then we assign the negative samples within the hypersphere into the HE group. Then the farthest $\beta$ proportion of the negative samples is assigned to the HU group, where $\beta \in (0, 1)$ is another hyperparameter. The remaining negative samples go to the UN group.

Our treatment assignment method aims to encapsulate $\alpha$ proportion of positive samples within a hypersphere defined by its center $a$ and radius $r$. The optimization problem can be formulated as:

$$\min_{r,a,\epsilon} F(r, a) = r^2 + C \sum_{(u,i) \in \mathcal{D}_{DP}} \epsilon_{u,i},$$

$$\text{s.t.} \quad \|x_{u,i} - a\|^2 \leq r^2 + \epsilon_{u,i}, \epsilon_{u,i} \geq 0, \forall (u, i) \in \mathcal{D}_{DP},$$

where $r$ is the radius of the hypersphere, $a$ is its center, and $\epsilon_{u,i}$ are slack variables that allow certain user-item samples to lie outside the hypersphere.

To solve this optimization problem, we incorporate the Lagrange multipliers $\lambda_{u,i} \geq 0$ and $\gamma_{u,i} \geq 0$ for each constraint:

$$L = r^2 + C \sum_{(u,i) \in \mathcal{D}_{DP}} \left\{ \epsilon_{u,i} - \lambda_{u,i} \left( r^2 + \epsilon_{u,i} - (x_{u,i}^2 - 2ax_{u,i} + a^2) \right) - \gamma_{u,i} \epsilon_{u,i} \right\}.$$

Then, the dual optimization problem can be represented as:

$$\max_{\lambda} \sum_{(u,i) \in \mathcal{D}_{DP}} \lambda_{u,i}(x_{u,i} \cdot x_{u,i}) - \sum_{(u,i),(u',i') \in \mathcal{D}_{DP}} \lambda_{u,i} \lambda_{u',i'}(x_{u,i} \cdot x_{u',i'}),$$

$$\text{s.t.} \quad 0 \leq \lambda_{u,i} \leq C.$$

To handle non-linearity in the data, we apply a kernel function $K(x_{u,i}, x_{u',i'})$, which implicitly maps the input data into a higher-dimensional feature space. A commonly used kernel is the Gaussian radial basis function (RBF) kernel, defined as:

$$K(x_{u,i}, x_{u',i'}) = \exp(-q\|x_{u,i} - x_{u',i'}\|),$$

where $q$ controls the width of the kernel and thus the smoothness of the decision boundary.

The squared distance of a sample $x$ from the center of the hypersphere is computed as:

$$d^2(x) = K(x, x) - 2 \sum_{(u,i) \in \mathcal{D}_{DP}} \lambda_{u',i'} K(x_{u,i}, x) + \sum_{(u,i),(u',i') \in \mathcal{D}_{DP}} \lambda_{u,i} \lambda_{u',i'} K(x_{u,i}, x_{u',i'}).$$

And the squared radius of the hypersphere is $d^2(x_v)$. where $x_v$ is a support vector.

Then, we find the smallest $\alpha$ and the largest $\beta$ proportion of $\{d^2(x_{u,i}) \mid (u, i) \in \mathcal{D}_0\}$, and denote the threshold as $r_1$ and $r_2$ respectively to classify $\mathcal{D}_0$ into the following groups:

$$(u, i) \in \mathcal{D}_{HE} \quad \text{if} \quad d^2(x_{u,i}) \leq r_1,$$
$$(u, i) \in \mathcal{D}_{UN} \quad \text{if} \quad r_1 < d^2(x_{u,i}) \leq r_2,$$
$$(u, i) \in \mathcal{D}_{HU} \quad \text{if} \quad d^2(x_{u,i}) > r_2.$$

### 3.4 Counterfactual Representations

After obtaining a stratification for the samples, we propose a causal representation learning method to predict the relevance between users and items across $\mathcal{D}$. When $O_{u,i} = 1$, indicating that the relevance scores for samples in HE and DP groups are observed, we can use pointwise loss to train the model based on the outcomes $Y(1)$. This ensures that the model effectively fits the potential outcome. In contrast, when the samples are not assigned the treatment group (i.e., $O_{u,i} = 0$, corresponding to the UN and HU groups), we treat these unobserved interactions as counterfactual data and employ

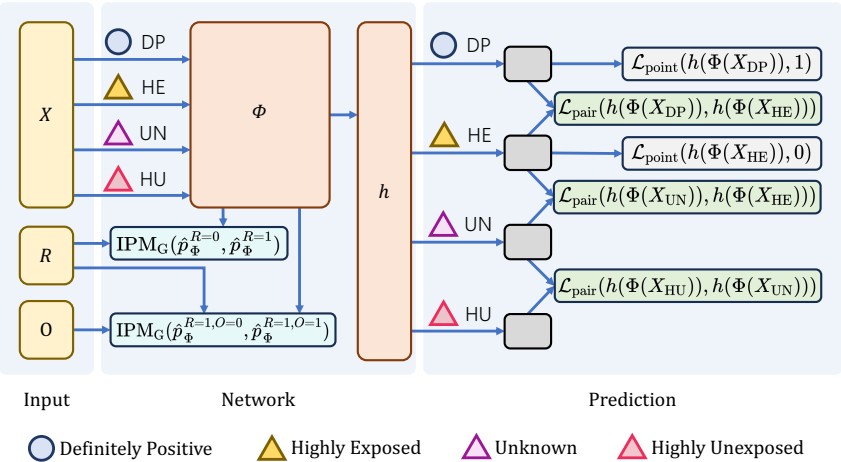

Figure 2: Representation learning framework through a counterfactual lens: exploring four strata of implicit feedback.

pairwise loss to model the relative rankings. This approach enhances the capability of our model to capture user-item relevance across all samples.

The user-item feature $x_{u,i}$ is transformed into the representation $\Phi(x_{u,i})$. The prediction model $h$ is then applied to the representation $\Phi(x_{u,i})$ to output the predicted interaction outcome $h(\Phi(x_{u,i}))$.

For DP and HE samples, we apply a pointwise cross-entropy loss. This ensures that the model accurately predicts the relevance based on the observed data. Specifically, for the DP samples, where $Y(1) = 1$, we have:

$$\mathcal{L}_{\text{point}}(h(\Phi(X_{DP})), 1) = -\frac{1}{|\mathcal{D}_{DP}|} \sum_{(u,i) \in \mathcal{D}_{DP}} \log h(\Phi(x_{u,i})).$$

While for the HE samples, where $Y(1) = 0$, the loss is defined as:

$$\mathcal{L}_{\text{point}}(h(\Phi(X_{HP})), 0) = -\frac{1}{|\mathcal{D}_{HE}|} \sum_{(u,i) \in \mathcal{D}_{HE}} \log(1 - h(\Phi(x_{u,i}))).$$

Therefore, the whole point loss is defined below:

$$\mathcal{L}_1 = \mathcal{L}_{\text{point}}(h(\Phi(X_{DP})), 1) + \mathcal{L}_{\text{point}}(h(\Phi(X_{HP})), 0).$$

For the unobserved interactions (i.e., $O_{u,i} = 0$, corresponding to $X_{UN}$ and $X_{HU}$), we use the pairwise loss to model the ranking of the interaction outcomes. Specifically, the following pairwise loss is applied to optimize the relative ranking between strata:

$$\mathcal{L}_{\text{pair}}(h(\Phi(X_+)), h(\Phi(X_-))) = \frac{1}{|\mathcal{D}_+| \cdot |\mathcal{D}_-|} \sum_{(u,i) \in \mathcal{D}_+, (j,k) \in \mathcal{D}_-} \log(\sigma(h(\Phi(x_{u,i}))) - h(\Phi(x_{j,k})),$$

where $\mathcal{D}_+$ and $\mathcal{D}_-$ are the sets of positive samples and negative samples. The pairwise ranking loss encourages the model to assign higher scores to positive samples over negative samples.

Specifically, we observe that the interaction probability for the HE group is significantly greater than that for the UN group, while the interaction probability for the HU group is markedly lower than that for the UN group. We apply the interaction probabilities among various user-item groups to establish positive and negative sample pairs and obtain the counterfactual loss:

$$\mathcal{L}_2 = \mathcal{L}_{\text{pair}}(h(\Phi(X_{DP})), h(\Phi(X_{HE})) + \mathcal{L}_{\text{pair}}(h(\Phi(X_{UN})), h(\Phi(X_{HE})) + \mathcal{L}_{\text{pair}}(h(\Phi(X_{HU})), h(\Phi(X_{UN})).$$

To mitigate distributional shifts between different strata of samples, we introduce Integral Probability Metric (IPM) regularization. IPM ensures that the representations of samples with confidence ($R = 1$) and samples that are not sure ($R = 0$) are aligned in the representation space. Furthermore, we also

introduce regularization for $R = 1, O = 0$ and $R = 1, O = 1$ samples to ensure consistency. The IPM regularization terms are defined as:

$$\mathcal{L}_{\text{IPM}} = \text{IPM}_G(p_\Phi^{R=0}, p_\Phi^{R=1}) + \text{IPM}_G(p_\Phi^{R=1,O=0}, p_\Phi^{R=1,O=1}).$$

By introducing IPM, we aim to ensure that the model learns representations that generalize well across different sample distributions. The IPM is defined as:

$$\text{IPM}_G(p, q) := \sup_{g \in G} \left| \int_S g(s)(p(s) - q(s)) \, ds \right|,$$

where $p$ and $q$ are two probability distributions, and $G$ is a class of functions for which we seek to optimize the difference in expectations.

The total loss for the model incorporates both pointwise and pairwise losses, addressing the observed and unobserved interactions across different strata. The final relevance prediction model is trained by minimizing the following loss:

$$\mathcal{L}_{\text{total}} = \lambda_{\text{point}} \mathcal{L}_1 + \lambda_{\text{pair}} \mathcal{L}_2 + \mathcal{L}_{\text{IPM}}.$$

### 3.5 Theoretical Analysis

We theoretically derive the generalization bound under our framework and show that minimizing the proposed pointwise loss $\mathcal{L}_1$, pairwise loss $\mathcal{L}_2$, and IPM loss $\mathcal{L}_{\text{IPM}}$ can effectively control the bound. First, we introduce the following assumption to ensure the existence of the inverse representation:

**Assumption 3.1** (Inverse Representation and Function Class [35]). *The representation $\Phi : \mathcal{X} \to \mathcal{A}$ is a one-to-one function, with inverse $\Psi$. Let G be a family of functions $g : \mathcal{A} \to \mathcal{Y}$. Assume there exists a constant $B_\Phi > 0$, such that $\frac{1}{B_\Phi} \cdot (h \circ \Phi \circ \Psi(a) - Y(1))^2 \in G$.*

Based on Assumption 3.1, we then derive the following generalization bound:

**Theorem 3.2** (Generalization Bound). *Under Assumption 3.1, the deviation between the estimated relevance $h(\Phi(x))$ and expected relevance $m_1(x) = \mathbb{E}[Y(1) \mid X = x]$ averaging on all user-item pairs has the upper bound:*

$$\mathbb{E}_x[(h(\Phi(x)) - m_1(x))^2] \leq \underbrace{\mathbb{E}_{x|r,o}[(h(\Phi(x)) - Y(1))^2 \mid R = 1, O = 1]}_{\text{factual loss of the DP and HE groups}}$$

$$+ \mathbb{P}(O = 0 \mid R = 1) \cdot B_\Phi \cdot \underbrace{IPM_G(p_\Phi^{R=1,O=0}, p_\Phi^{R=1,O=1})}_{\text{distribution shift on O given R=1}} + \underbrace{\mathbb{P}(R = 0)}_{\text{UN group}} \cdot B_\Phi \cdot \underbrace{IPM_G(p_\Phi^{R=0}, p_\Phi^{R=1})}_{\text{distribution shift on R}}$$

$$- \underbrace{\mathbb{E}[(Y(1) - m_1(x))^2]}_{\text{variance of potential outcome}}.$$

In the above bound, the first term is the factual loss based on the true value of $Y(1)$ of the DP and HE groups, which can be controlled by minimizing the $\mathcal{L}_{\text{point}}$ and $\mathcal{L}_{\text{pair}}$. The second and third terms are the IPM distance measuring the distribution shift on $O = 1$ and $O = 0$ given $R = 1$ group and distribution shift on $R = 1$ and $R = 0$ weighted by the proportion of HU group given $R = 1$ and proportion of UN group $\mathbb{P}(R = 0)$, respectively. These IPM distance terms can also be effectively controlled by minimizing the proposed $\mathcal{L}_{\text{IPM}}$. Intuitively, if $\mathbb{P}(R = 0) = 0$, there is no need to control the distribution shift on $R$. The last term measures the minimal variance of potential outcomes, which is independent of the model selection. See Appendix A for the detailed proof.

## 4 Experiments

### 4.1 Experimental Setup

**Datasets.** To evaluate the performance of unbiased recommendations, we utilize two real-world datasets: **Coat** and **Yahoo! R3**. Each dataset includes both biased training data and an unbiased test set. The **Coat** dataset contains 6,960 biased ratings and 4,640 unbiased ratings provided by 290 users for 300 items, where each user rates 16 randomly selected items. The **Yahoo! R3** dataset includes

Table 2: Ranking performance on Yahoo and Coat. We bold the best results and underline the best baseline. The results with * indicate p < 0.05 using the pairwise t-test with the best competitor.

| | Methods | K=3 | | | K=5 | | | K=8 | | |
|---|---|---|---|---|---|---|---|---|---|---|
| | | NDCG@K | Recall@K | MAP@K | NDCG@K | Recall@K | MAP@K | NDCG@K | Recall@K | MAP@K |
| Yahoo | ExpoMF | $0.524 \pm 0.008$ | $0.581 \pm 0.012$ | $0.461 \pm 0.008$ | $0.588 \pm 0.007$ | $0.736 \pm 0.010$ | $0.509 \pm 0.007$ | $0.652 \pm 0.005$ | $0.912 \pm 0.003$ | $0.548 \pm 0.006$ |
| | WMF | $0.538 \pm 0.005$ | $0.596 \pm 0.006$ | $0.470 \pm 0.004$ | $0.600 \pm 0.005$ | $0.755 \pm 0.011$ | $0.529 \pm 0.004$ | $0.663 \pm 0.004$ | $0.918 \pm 0.004$ | $0.561 \pm 0.004$ |
| | Rel-MF | $0.534 \pm 0.008$ | $0.599 \pm 0.009$ | $0.465 \pm 0.008$ | $0.593 \pm 0.007$ | $0.749 \pm 0.010$ | $0.523 \pm 0.007$ | $0.653 \pm 0.004$ | $0.918 \pm 0.003$ | $0.555 \pm 0.005$ |
| | Rel-MF-du | $0.540 \pm 0.008$ | $0.596 \pm 0.008$ | $0.478 \pm 0.009$ | $\underline{0.611 \pm 0.007}$ | $0.756 \pm 0.010$ | $0.530 \pm 0.009$ | $0.668 \pm 0.007$ | $0.915 \pm 0.009$ | $0.557 \pm 0.008$ |
| | BPR | $0.517 \pm 0.003$ | $0.574 \pm 0.005$ | $0.455 \pm 0.003$ | $0.581 \pm 0.006$ | $0.732 \pm 0.011$ | $0.502 \pm 0.005$ | $0.654 \pm 0.004$ | $0.905 \pm 0.005$ | $0.542 \pm 0.004$ |
| | UBPR | $0.532 \pm 0.005$ | $0.592 \pm 0.005$ | $0.470 \pm 0.005$ | $0.596 \pm 0.002$ | $0.746 \pm 0.007$ | $0.517 \pm 0.003$ | $0.657 \pm 0.002$ | $0.913 \pm 0.003$ | $0.555 \pm 0.004$ |
| | UBPR-nclip | $0.536 \pm 0.008$ | $0.597 \pm 0.008$ | $0.474 \pm 0.009$ | $0.597 \pm 0.006$ | $0.746 \pm 0.010$ | $0.522 \pm 0.009$ | $0.659 \pm 0.007$ | $0.914 \pm 0.007$ | $0.557 \pm 0.008$ |
| | UPL | $\underline{0.546 \pm 0.005}$ | $\underline{0.603 \pm 0.011}$ | $0.483 \pm 0.004$ | $0.610 \pm 0.004$ | $\underline{0.759 \pm 0.007}$ | $0.532 \pm 0.005$ | $0.668 \pm 0.004$ | $0.922 \pm 0.007$ | $0.568 \pm 0.004$ |
| | RecRec | $0.545 \pm 0.007$ | $0.602 \pm 0.004$ | $\underline{0.484 \pm 0.008}$ | $0.607 \pm 0.004$ | $0.757 \pm 0.004$ | $\underline{0.533 \pm 0.006}$ | $\underline{0.669 \pm 0.004}$ | $\underline{0.923 \pm 0.008}$ | $\underline{0.570 \pm 0.006}$ |
| | Ours | $\mathbf{0.562^* \pm 0.007}$ | $\mathbf{0.624^* \pm 0.009}$ | $\mathbf{0.499^* \pm 0.007}$ | $\mathbf{0.625^* \pm 0.005}$ | $\mathbf{0.776^* \pm 0.008}$ | $\mathbf{0.547^* \pm 0.006}$ | $\mathbf{0.681^* \pm 0.004}$ | $\mathbf{0.930 \pm 0.004}$ | $\mathbf{0.582^* \pm 0.005}$ |
| Coat | ExpoMF | $0.324 \pm 0.007$ | $0.340 \pm 0.010$ | $0.256 \pm 0.007$ | $0.372 \pm 0.006$ | $0.459 \pm 0.012$ | $0.287 \pm 0.007$ | $0.428 \pm 0.008$ | $0.601 \pm 0.008$ | $0.315 \pm 0.008$ |
| | WMF | $0.333 \pm 0.013$ | $0.322 \pm 0.019$ | $0.264 \pm 0.015$ | $0.369 \pm 0.016$ | $0.412 \pm 0.020$ | $0.294 \pm 0.015$ | $0.426 \pm 0.016$ | $0.610 \pm 0.025$ | $0.325 \pm 0.015$ |
| | RelMF | $0.338 \pm 0.013$ | $0.344 \pm 0.018$ | $0.267 \pm 0.010$ | $0.385 \pm 0.005$ | $\underline{0.462 \pm 0.012}$ | $0.304 \pm 0.006$ | $0.433 \pm 0.008$ | $0.614 \pm 0.022$ | $0.338 \pm 0.006$ |
| | RelMF-du | $0.340 \pm 0.016$ | $0.332 \pm 0.023$ | $0.275 \pm 0.013$ | $0.374 \pm 0.010$ | $0.420 \pm 0.014$ | $0.307 \pm 0.009$ | $\underline{0.458 \pm 0.015}$ | $\underline{0.640 \pm 0.021}$ | $0.353 \pm 0.012$ |
| | BPR | $0.324 \pm 0.011$ | $0.325 \pm 0.018$ | $0.265 \pm 0.010$ | $0.370 \pm 0.010$ | $0.433 \pm 0.017$ | $0.290 \pm 0.008$ | $0.445 \pm 0.009$ | $0.640 \pm 0.017$ | $0.335 \pm 0.007$ |
| | UBPR | $0.343 \pm 0.012$ | $0.342 \pm 0.018$ | $0.269 \pm 0.012$ | $0.384 \pm 0.009$ | $0.451 \pm 0.017$ | $0.306 \pm 0.009$ | $0.449 \pm 0.009$ | $0.642 \pm 0.023$ | $0.339 \pm 0.008$ |
| | UBPR-nclip | $0.335 \pm 0.007$ | $0.345 \pm 0.013$ | $0.261 \pm 0.006$ | $0.368 \pm 0.011$ | $0.430 \pm 0.022$ | $0.290 \pm 0.009$ | $0.445 \pm 0.007$ | $0.640 \pm 0.012$ | $0.335 \pm 0.006$ |
| | UPL-BPR | $0.345 \pm 0.009$ | $0.343 \pm 0.014$ | $0.273 \pm 0.008$ | $0.377 \pm 0.009$ | $0.427 \pm 0.025$ | $0.302 \pm 0.007$ | $0.438 \pm 0.009$ | $0.615 \pm 0.014$ | $0.340 \pm 0.008$ |
| | RecRec | $\underline{0.360 \pm 0.008}$ | $\underline{0.365 \pm 0.014}$ | $\underline{0.284 \pm 0.005}$ | $\underline{0.392 \pm 0.009}$ | $0.452 \pm 0.019$ | $\underline{0.314 \pm 0.006}$ | $0.454 \pm 0.007$ | $0.629 \pm 0.018$ | $\underline{0.354 \pm 0.004}$ |
| | Ours | $\mathbf{0.368 \pm 0.011}$ | $\mathbf{0.382^* \pm 0.012}$ | $\mathbf{0.296^* \pm 0.009}$ | $\mathbf{0.414^* \pm 0.012}$ | $\mathbf{0.478 \pm 0.014}$ | $\mathbf{0.332^* \pm 0.009}$ | $\mathbf{0.473^* \pm 0.012}$ | $\mathbf{0.660 \pm 0.021}$ | $\mathbf{0.369^* \pm 0.009}$ |

Table 3: Ablation study on the Yahoo and Coat datasets.

| | Methods | K=3 | | | K=5 | | | K=8 | | |
|---|---|---|---|---|---|---|---|---|---|---|
| | | NDCG@K | Recall@K | MAP@K | NDCG@K | Recall@K | MAP@K | NDCG@K | Recall@K | MAP@K |
| Yahoo | w/o Wass w/o Pair | $0.546 \pm 0.008$ | $0.609 \pm 0.009$ | $0.483 \pm 0.009$ | $0.611 \pm 0.007$ | $0.765 \pm 0.008$ | $0.531 \pm 0.009$ | $0.668 \pm 0.007$ | $0.923 \pm 0.009$ | $0.566 \pm 0.004$ |
| | w/o Wass w/o Point | $0.552 \pm 0.007$ | $0.615 \pm 0.010$ | $0.488 \pm 0.006$ | $0.616 \pm 0.008$ | $0.770 \pm 0.008$ | $0.536 \pm 0.007$ | $0.673 \pm 0.006$ | $0.927 \pm 0.005$ | $0.572 \pm 0.005$ |
| | w/o Pair | $0.552 \pm 0.008$ | $0.615 \pm 0.010$ | $0.488 \pm 0.008$ | $0.616 \pm 0.006$ | $0.770 \pm 0.009$ | $0.536 \pm 0.007$ | $0.673 \pm 0.007$ | $0.927 \pm 0.007$ | $0.572 \pm 0.006$ |
| | w/o Point | $0.554 \pm 0.005$ | $0.614 \pm 0.007$ | $0.491 \pm 0.006$ | $0.619 \pm 0.007$ | $0.774 \pm 0.004$ | $0.540 \pm 0.007$ | $0.675 \pm 0.008$ | $0.927 \pm 0.006$ | $0.575 \pm 0.007$ |
| | w/o Wass | $0.558 \pm 0.008$ | $0.623 \pm 0.010$ | $0.494 \pm 0.008$ | $0.620 \pm 0.009$ | $0.771 \pm 0.009$ | $0.540 \pm 0.008$ | $0.676 \pm 0.006$ | $0.927 \pm 0.007$ | $0.576 \pm 0.008$ |
| | **All** | $\mathbf{0.562 \pm 0.007}$ | $\mathbf{0.624 \pm 0.009}$ | $\mathbf{0.499 \pm 0.007}$ | $\mathbf{0.625 \pm 0.005}$ | $\mathbf{0.776 \pm 0.008}$ | $\mathbf{0.547 \pm 0.006}$ | $\mathbf{0.681 \pm 0.004}$ | $\mathbf{0.930 \pm 0.004}$ | $\mathbf{0.582 \pm 0.005}$ |
| Coat | w/o Wass w/o Pair | $0.362 \pm 0.010$ | $0.376 \pm 0.010$ | $0.280 \pm 0.009$ | $0.398 \pm 0.011$ | $0.477 \pm 0.012$ | $0.314 \pm 0.008$ | $0.450 \pm 0.010$ | $0.618 \pm 0.019$ | $0.347 \pm 0.007$ |
| | w/o Wass w/o Point | $0.363 \pm 0.009$ | $0.378 \pm 0.010$ | $0.285 \pm 0.009$ | $0.399 \pm 0.010$ | $0.477 \pm 0.013$ | $0.319 \pm 0.009$ | $0.450 \pm 0.011$ | $0.609 \pm 0.020$ | $0.352 \pm 0.009$ |
| | w/o Pair | $0.366 \pm 0.011$ | $0.366 \pm 0.012$ | $0.290 \pm 0.008$ | $0.412 \pm 0.010$ | $\mathbf{0.485 \pm 0.015}$ | $0.327 \pm 0.011$ | $0.472 \pm 0.013$ | $0.646 \pm 0.022$ | $0.368 \pm 0.009$ |
| | w/o Point | $0.361 \pm 0.010$ | $0.363 \pm 0.010$ | $0.289 \pm 0.009$ | $0.407 \pm 0.010$ | $0.484 \pm 0.013$ | $0.325 \pm 0.009$ | $0.455 \pm 0.011$ | $0.614 \pm 0.021$ | $0.356 \pm 0.009$ |
| | w/o Wass | $0.364 \pm 0.010$ | $0.377 \pm 0.011$ | $0.281 \pm 0.010$ | $0.402 \pm 0.012$ | $0.478 \pm 0.014$ | $0.317 \pm 0.010$ | $0.468 \pm 0.012$ | $0.656 \pm 0.020$ | $0.357 \pm 0.008$ |
| | **All** | $\mathbf{0.368 \pm 0.011}$ | $\mathbf{0.382 \pm 0.012}$ | $\mathbf{0.296 \pm 0.009}$ | $\mathbf{0.414 \pm 0.012}$ | $0.478 \pm 0.014$ | $\mathbf{0.332 \pm 0.009}$ | $\mathbf{0.473 \pm 0.012}$ | $\mathbf{0.660 \pm 0.021}$ | $\mathbf{0.369 \pm 0.009}$ |

311,704 biased ratings and 54,000 unbiased ratings from 15,400 users interacting with 1,000 items. The positive samples are sparse, which is consistent with the real-world situation. We employed the preprocessing steps following previous studies [23, 36], which can be seen in the Appendix B.

**Evaluation Metrics and Details.** We use three common metrics to evaluate implicit recommendation systems: NDCG@k (Normalized Discounted Cumulative Gain), Recall@k, and MAP@k (Mean Average Precision). DCG@k evaluates the ranking quality by giving more weight to relevant items appearing earlier in the list. Recall@k measures how many relevant items are retrieved within the top $k$ recommendations. MAP@k calculates the mean precision across users, considering both relevance and order. Results are presented for $k = 3$, $k = 5$, and $k = 8$.

**Hyperparamter Tuning.** For each dataset, the data was divided into training and test sets. A portion of 10% from the training set was randomly selected to serve as the validation set for hyperparameter tuning. Several key parameters were adjusted during this phase. The latent factor dimensions, representing user-item interactions, were explored within the range of 100 to 300, while the L2 regularization term was fine-tuned between $[10^{-7}, 10^{-3}]$ for all models, and the $\lambda_{\text{point}}$ as well as $\lambda_{\text{pair}}$ are tuned in $\{0.01, 0.1, 1, 10, 100\}$.

**Baselines.** To achieve a comprehensive comparison, we consider the following methods as baselines: **WMF** [1], **ExpoMF** [8], **Rel-MF** [16], **Rel-MF-du** [16], **BPR** [13], **UBPR** [23], **UBPR-nclip** [23], **UPL** [37], and **RecRec** [36].

### 4.2 Performance Comparison

We evaluate the performance of our proposed method against several baseline approaches on multiple datasets, as shown in Table 2. The results highlight several key findings. Traditional methods like BPR and WMF show moderate performance but struggle with the PU and MNAR challenges in implicit feedback. BPR misclassifies potential positives as negatives by assuming unobserved interactions are negative, while WMF treats unobserved interactions as low-weight positives, failing to fully address false negatives. Methods like ExpoMF and Rel-MF improve by modeling exposure and item popularity, reducing bias. ExpoMF incorporates exposure variables but remains limited by traditional matrix factorization, while Rel-MF leverages item popularity to estimate propensities, though it still

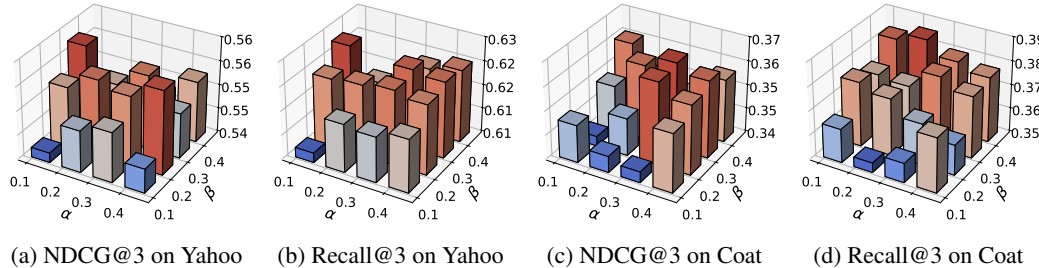

(a) NDCG@3 on Yahoo     (b) Recall@3 on Yahoo     (c) NDCG@3 on Coat     (d) Recall@3 on Coat

Figure 3: Sensitivity analysis on the proportion of the HE group $\alpha$ and the proportion of the HU group $\beta$ for samples in $\mathcal{D}_0$.

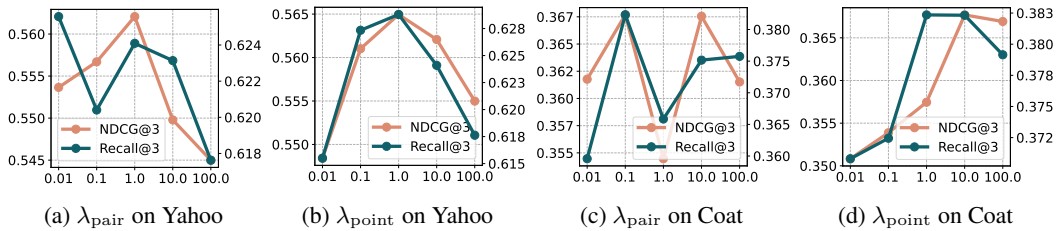

(a) $\lambda_{\text{pair}}$ on Yahoo     (b) $\lambda_{\text{point}}$ on Yahoo     (c) $\lambda_{\text{pair}}$ on Coat     (d) $\lambda_{\text{point}}$ on Coat

Figure 4: The sensitivity of balancing weight between pointwise and pairwise losses.

struggles with unobserved items. UBPR and UBPR-nclip extend BPR to address PU and MNAR problems, with UBPR-nclip further reducing bias through a non-clipping estimator. UPL simplifies pairwise learning, performing well in high-variance scenarios. Finally, ReCRec shows strong results but is computationally intensive, limiting scalability. Despite improvements over simpler models, these baselines still fail to fully address the challenges of implicit feedback. Our method outperforms all baselines by explicitly modeling exposure and preference to address PU and MNAR issues. By combining the proposed treatment imputation and balanced representation learning, our method leads to more accurate predictions.

### 4.3 Ablation Study

We conduct ablation studies to validate the effectiveness of our model by examining the impact of the Wasserstein distance (Wass), pairwise loss (Pair), and pointwise loss (Point). Table 3 shows performance metrics (NDCG@K, Recall@K, MAP@K) on Yahoo and Coat datasets. Removing any component degrades performance, as each addresses positive-unlabeled and MNAR challenges in implicit feedback. Removing both Wass and Pair (w/o Wass w/o Pair) yields the lowest performance, as the model cannot stratify negative samples or learn robust representations. Similarly, removing Wass and Point (w/o Wass w/o Point) compromises the causal learning framework. Removing only Pair (w/o Pair) or Point (w/o Point) also reduces performance, though less severely. Both losses contribute uniquely: Pair improves ranking, while Point predicts click likelihood under exposure. Removing only Wass (w/o Wass) degrades performance, as the model loses the ability to stratify negative samples by exposure probability, crucial for distinguishing unclicked items due to low relevance versus lack of exposure. In summary, the proposed model with all components (Wass, Pair, Point) achieves the best performance, highlighting the importance of each in addressing implicit feedback challenges and estimating counterfactual outcomes with missing treatments.

### 4.4 Sensitivity Analysis

**Threshold and proportion.** Figure 3 shows the sensitivity of the model's performance on the proportion of the HE group $\alpha$ and the proportion of the HU group $\beta$ for samples in $\mathcal{D}_0$. With very low or very high $\alpha$, performance tends to degrade because the model potentially excludes relevant samples or includes irrelevant samples. Similarly, an extreme $\beta$ degrades the performance.

**Coefficients of the loss function.** As shown in Figure 4, we investigate the impact of the coefficients $\lambda_{\text{pair}}$ and $\lambda_{\text{point}}$ for the pairwise and pointwise losses in the proposed model. We evaluate the effects of varying these coefficients on NDCG@K, Recall@K, and MAP@K across the Yahoo! R3 and Coat datasets. The results show that setting $\lambda_{\text{pair}}$ and $\lambda_{\text{point}}$ within a moderate range (e.g., 0.1, 1, or 10)

leads to significant improvements in ranking accuracy and click prediction. Excessively large values (e.g., 100) overemphasize either ranking or click prediction, degrading performance. Similarly, very small values (e.g., 0.01) fail to leverage both losses effectively, resulting in poor performance. The optimal performance is achieved when these coefficients are set to 1 or 10, striking a balance between ranking accuracy and click likelihood prediction.

# 5 Related Work

## 5.1 Implicit Feedback

Early work solving the positive-unlabeled problems in implicit recommender systems includes weighted matrix factorization (WMF), and some approaches like MF [1] downweight negative samples uniformly, some reweight samples via user activities [38], and others use item popularity to adjust weights [39]. Exposure models, such as ExpoMF [8] models exposure probabilities based on item popularity and text topics, while other methods incorporate social or community data [12, 40]. Most methods rely on pointwise loss, but pairwise loss (e.g., Bayesian personalized ranking, BPR [13]) is better suited for ranking tasks by learning relative preferences. However, these methods do not consider the implicit feedback in MNAR, thus may get biased relevance prediction.

Recently, there are some propensity-based methods proposed to address positive-unlabeled and MNAR issues. Rel-MF [22] leverages item popularity for propensity estimation and solves MNAR problem in implicit feedback, while joint learning approaches [41, 42] infer both propensity and recommendation models. Methods using small unbiased datasets employ embedding alignment [6], knowledge distillation [43], or meta-learning [44] to learn exposure/propensity models. For pairwise loss, UBPR [23] proposed a debiased loss. However, the propensity models are prone to be overly confident, generating extremely inaccurate propensity score estimation [25, 26, 45]. And the bias and variance of the estimator can be extremely large with small propensity [27, 28].

## 5.2 Causal Recommendation

Causal recommendations refer to applying the causal frameworks [46, 47, 48] to predict the conversion rates [49] and provide personalized recommendations [50, 51]. Compared with previous recommendation methods, the causal recommendations can address various biases, including the confounding bias [52, 53, 54], popularity bias [55], selection bias [18, 56, 57], and exposure bias [58], which previous methods fail to address because they highly rely on the associations. Most of the methods are based on inverse-propensity or doubly robust weighting [59, 60, 61]. Some studies propose to use the same training and inference space with entire-space multi-task learning approaches [62, 63, 28, 64]. They jointly train the parameters to achieve better recommendation performance. More recent works focus on counterfactual learning, which predicts the counterfactual outcomes for the user-item pairs [65, 66, 67]. However, most of them cannot achieve individual-level counterfactual predictions. [68] extends the above methods to perform individual counterfactual predictions. However, the above methods are mainly applied in explicit feedback recommendations. It is impossible to directly apply the above methods to the implicit feedback because it is challenging to determine whether an observed negative user-item pair is irrelevant in implicit feedback recommender systems.

# 6 Conclusion

In this paper, we focus on the problem of inferring the true relevance or preference of a user in the implicit feedback recommendation scenarios. Specifically, to the best of our knowledge, we are the first paper to formalize the relevance prediction problem as a counterfactual outcome estimation problem with missing treatments, which provides a novel approach to tackle this problem. Correspondingly, we propose a sample stratification method, which uses a treatment variable imputation method with feature similarity-based confidence, reflecting different mechanisms of negative sample generation. In addition, we propose a balanced representation-based causal learning framework to answer the formalized counterfactual questions and theoretically derive the generalization bound of our causal learning model, showing that minimizing the proposed loss functions can effectively control the bound. Extensive experiments on public benchmark datasets show the effectiveness of our method. One potential limitation is that the method requires pre-specification of the distance thresholds. Changing the framework to an end-to-end manner may further help improve performance.

## Acknowledgments and Disclosure of Funding

The authors thank the anonymous reviewers for their valuable comments. HL was supported by the National Natural Science Foundation of China (623B2002). MG was supported by ARC DP240102088 and WIS-MBZUAI 142571.

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

# A Proof of Theorem

**Assumption A.1** (Inverse Representation and Function Class [35]). The representation $\Phi : \mathcal{X} \to \mathcal{A}$ is a one-to-one function, with inverse $\Psi$. Let G be a family of functions $g : \mathcal{A} \to \mathcal{Y}$. Assume there exists a constant $B_\Phi > 0$, such that $\frac{1}{B_\Phi} \cdot (h \circ \Phi \circ \Psi(a) - Y(1))^2 \in G$.

Based on Assumption A.1, we then derive the following generalization bound:

**Theorem A.2** (Generalization Bound). *Under Assumption A.1, the deviation between the estimated relevance $h(\Phi(x))$ and expected relevance $m_1(x) = \mathbb{E}[Y(1) \mid X = x]$ averaging on all user-item pairs has the upper bound:*

$$\mathbb{E}_x[(h(\Phi(x)) - m_1(x))^2] \leq \underbrace{\mathbb{E}_{x|r,o}[(h(\Phi(x)) - Y(1))^2 \mid R = 1, O = 1]}_{\text{factual loss of the DP and HE groups}}$$

$$+ \mathbb{P}(O = 0 \mid R = 1) \cdot B_\Phi \cdot \underbrace{IPM_G(p_\Phi^{R=1,O=0}, p_\Phi^{R=1,O=1})}_{\text{distribution shift on O given R=1}} + \underbrace{\mathbb{P}(R = 0)}_{\text{UN group}} \cdot B_\Phi \cdot \underbrace{IPM_G(p_\Phi^{R=0}, p_\Phi^{R=1})}_{\text{distribution shift on R}}$$

$$- \underbrace{\mathbb{E}[(Y(1) - m_1(x))^2]}_{\text{variance of potential outcome}}.$$

*Proof.* For simplicity, we define $\epsilon(x) = \mathbb{E}[(h(\Phi(x)) - Y(1))^2 \mid X = x]$ and $\epsilon_{Total} = \mathbb{E}_x[\epsilon(x)] = \mathbb{E}[(h(\Phi(x)) - Y(1))^2]$. The target error that we would like to bound is $\epsilon_{Target} := \mathbb{E}_x[(h(\Phi(x)) - m_1(x))^2]$. We will first derive the connection between the target error $\epsilon_{Target}$ and the total error $\epsilon_{Total}$. Consider the following conditional expectation:

$$\mathbb{E}[(h(\Phi(x)) - Y(1))^2 \mid X = x]$$
$$= \mathbb{E}[((h(\Phi(x)) - m_1(x)) + (m_1(x) - Y(1)))^2 \mid X = x]$$
$$= \mathbb{E}[(h(\Phi(x)) - m_1(x))^2 \mid X = x] + \mathbb{E}[(m_1(x) - Y(1))^2 \mid X = x]$$
$$+ 2\mathbb{E}[(h(\Phi(x)) - m_1(x)) \cdot (m_1(x) - Y(1)) \mid X = x]$$
$$= \mathbb{E}[(h(\Phi(x)) - m_1(x))^2 \mid X = x] + \mathbb{E}[(m_1(x) - Y(1))^2 \mid X = x].$$

Taking expectation w.r.t the distribution of $X$, we have:

$$\epsilon_{Total}$$
$$= \mathbb{E}_x\{\mathbb{E}[(h(\Phi(x)) - Y(1))^2 \mid X = x]\}$$
$$= \mathbb{E}_x\{\mathbb{E}[(h(\Phi(x)) - m_1(x))^2 \mid X = x]\} + \mathbb{E}_x\{\mathbb{E}[(m_1(x) - Y(1))^2 \mid X = x]\}$$
$$= \mathbb{E}[(h(\Phi(x)) - m_1(x))^2] + \mathbb{E}[(m_1(x) - Y(1))^2]$$
$$= \epsilon_{Target} + \mathbb{E}[(m_1(x) - Y(1))^2]. \tag{2}$$

Denoting $L(x) = (h(\Phi(x)) - Y(1))^2$, $v_0 = P(R = 0)$, $\epsilon^{R=1} = \mathbb{E}_x[\epsilon(x) \mid R = 1]$ and $\epsilon^{R=0} = \mathbb{E}_x[\epsilon(x) \mid R = 0]$, we can decompose $\epsilon_{Total}$ as follows:

$$\epsilon_{Total}$$
$$= \mathbb{E}_{x|r}[\epsilon(x) \mid R = 1]P(R = 1) + \mathbb{E}_{x|r}[\epsilon(x) \mid R = 0]P(R = 0)$$
$$= \epsilon^{R=1}(1 - v_0) + \epsilon^{R=0}v_0$$
$$= \epsilon^{R=1} - \epsilon^{R=1}v_0 + \epsilon^{R=0}v_0$$
$$= \epsilon^{R=1} + v_0(\epsilon^{R=0} - \epsilon^{R=1})$$
$$= \epsilon^{R=1} + v_0 \left( \int \epsilon(x)p(x \mid R = 0)dx - \int \epsilon(x)p(x \mid R = 1)dx \right)$$
$$= \epsilon^{R=1} + v_0 \int \epsilon(x)(p(x \mid R = 0) - p(x \mid R = 1))dx$$
$$= \epsilon^{R=1} + v_0 \int \mathbb{E}[(h(\Phi(x)) - Y(1))^2 \mid X = x](p(x \mid R = 0) - p(x \mid R = 1))dx$$
$$= \epsilon^{R=1} + v_0 \int L(x)(p(x|R = 0) - p(x|R = 1))dx. \tag{3}$$

Based on Assumption A.1, we have $L(x)/B_\Phi = \frac{1}{B_\Phi}(h \circ \Psi \circ \Psi(a) - Y(1))^2 \in G$, and thus

$$\int L(x)(p(x|R=0) - p(x|R=1))dx$$

$$= B_\Phi \int \frac{L(x)}{B_\Phi}(p(x|R=0) - p(x|R=1))dx$$

$$= B_\Phi \int \frac{L(\Psi(a))}{B_\Phi}(p(a|R=0) - p(a|R=1))da$$

$$\leq B_\Phi \cdot \sup_{g \in G} |\int g(a)(p_\Phi^{R=0}(a) - p_\Phi^{R=1}(a))da|$$

$$= B_\Phi \cdot IPM_G(p_\Phi^{R=0}, p_\Phi^{R=1}). \tag{4}$$

Combining Eq. (3) and Eq. (4), we have

$$\epsilon_{Total} \leq \epsilon^{R=1} + v_0 \cdot B_\Phi \cdot IPM_G(p_\Phi^{R=0}, p_\Phi^{R=1}). \tag{5}$$

Next, we denote $u_0 = P(O = 0 \mid R = 1), \epsilon^{R=1,O=1} = \mathbb{E}_{x|r,o}[\epsilon(x) \mid R = 1, O = 1]$ and $\epsilon^{R=1,O=0} = \mathbb{E}_{x|r,o}[\epsilon(x) \mid R = 1, O = 0]$ and then decompose $\epsilon^{R=1}$ as follows:

$$\epsilon^{R=1}$$

$$= \epsilon^{R=1,O=1} \cdot (1 - u_0) + \epsilon^{R=1,O=0} \cdot u_0$$

$$= \epsilon^{R=1,O=1} + u_0 \cdot (\epsilon^{R=1,O=0} - \epsilon^{R=1,O=1})$$

$$= \epsilon^{R=1,O=1} + u_0 \left( \int \epsilon(x)p(x \mid R = 1, O = 0)dx - \int \epsilon(x)p(x \mid R = 1, O = 1)dx \right)$$

$$= \epsilon^{R=1,O=1} + u_0 \int \epsilon(x)(p(x \mid R = 1, O = 0) - p(x \mid R = 1, O = 1))dx$$

$$= \epsilon^{R=1,O=1}$$

$$+ u_0 \int \mathbb{E}[(h(\Phi(x)) - Y(1))^2 \mid X = x](p(x \mid R = 1, O = 0) - p(x \mid R = 1, O = 1))dx$$

$$= \epsilon^{R=1,O=1} + u_0 \int L(x)(p(x \mid R = 1, O = 0) - p(x \mid R = 1, O = 1))dx. \tag{6}$$

Analogous to the derivation of Eq. (4), we have

$$\int L(x)(p(x \mid R = 1, O = 0) - p(x \mid R = 1, O = 1))dx$$

$$= B_\Phi \int \frac{L(x)}{B_\Phi}(p(x \mid R = 1, O = 0) - p(x \mid R = 1, O = 1))dx$$

$$= B_\Phi \int \frac{L(\Psi(a))}{B_\Phi}(p(a \mid R = 1, O = 0) - p(a \mid R = 1, O = 1))da$$

$$\leq B_\Phi \cdot \sup_{g \in G} |\int g(a)(p_\Phi^{R=1,O=0}(a) - p_\Phi^{R=1,O=1}(a))da|$$

$$= B_\Phi \cdot IPM_G(p_\Phi^{R=1,O=0}, p_\Phi^{R=1,O=1}). \tag{7}$$

Combining Eq. (6) and Eq. (7), we have

$$\epsilon^{R=1} \leq \epsilon^{R=1,O=1} + u_0 \cdot B_\Phi \cdot IPM_G(p_\Phi^{R=1,O=0}, p_\Phi^{R=1,O=1}). \tag{8}$$

Note that $\epsilon^{R=1,O=1} = \mathbb{E}_{x|r,o}[\epsilon(x) \mid R = 1, O = 1] = \mathbb{E}_{x|r,o}[(h(\Phi(x)) - Y(1))^2 \mid R = 1, O = 1]$, and combining the results of Eq. (2), Eq. (5) and Eq. (8), we complete the proof of the theorem.

$$\square$$

## B  Data Preprocess

All of the datasets used in this paper contain user ratings on an item as explicit feedback. So we simulate the implicit feedback mechanism using the following data preprocessing pipeline.

1. Transform all ratings into relevance scores using the following formula:

$$\gamma_{u,i} = \epsilon + (1 - \epsilon)\frac{2r_{u,i} - 1}{2r_{\max} - 1},$$

   where $r_{u,i}$ denotes the rating for each user-item pair in the observed set $\mathcal{O}$, and $r_{\max}$ is the maximum rating. The parameter $\epsilon \in [0, 1]$ controls the noise level. Following the previous studies [23, 36], we set $\epsilon = 0.1$ for the training datasets and $\epsilon = 0$ for the test datasets to ensure unbiased evaluation.

2. Sample the binary relevance $S_{u,i}$ using Bernoulli sampling:

$$S_{u,i} \sim \text{Bern}(\gamma_{u,i}), \quad \forall (u, i) \in \mathcal{O},$$

   where $\text{Bern}(\cdot)$ denotes the Bernoulli distribution.

3. Define the exposure variable for all user-item pair $(u, i) \in \mathcal{U} \times \mathcal{I}$:

$$O_{u,i} = \begin{cases} 1 & \text{if item } i \text{ is rated by user } u, \\ 0 & \text{if item } i \text{ is not rated by user } u. \end{cases}$$

4. Finally, we sample the binary outcome as the implicit feedback:

$$Y_{u,i} = \begin{cases} S_{u,i} & \text{if } O_{u,i} = 1, \\ 0 & \text{if } O_{u,i} = 1. \end{cases}$$

   Note that $S_{u,i}$ and $O_{u,i}$ are unobservable in our setting and the training data is $\{(u, i, Y_{u,i}) : (u, i) \in \mathcal{U} \times \mathcal{I}\}$.

## C  Efficient Alternatives of Kernel-Based Hypersphere Model

Although the kernel-based hypersphere model in Section 3.3 introduces $O(m^2)$ complexity with $m$ positive samples, we would like to emphasize that the calculation of the kernel function is only one possible option; the core lies in the framework of treatment imputation with confidence. Here, we provide two more efficient and scalable kernel function approximation methods, i.e., Random Fourier Features (RFF) [69] and Nyström approximation [70] to replace the method in Section 3.3. Experimental results on the Coat and Yahoo datasets are presented in 4, showing that regardless of the kernel function calculation method adopted, the ranking performance is guaranteed.

Table 4: Ranking performance with efficient alternatives of kernel-based hypersphere model.

| Methods | K=3 | | | K=5 | | | K=8 | | |
|---|---|---|---|---|---|---|---|---|---|
| | NDCG@K | Recall@K | MAP@K | NDCG@K | Recall@K | MAP@K | NDCG@K | Recall@K | MAP@K |
| **Yahoo** Ours (SVDD) | $0.562 \pm 0.007$ | $0.624 \pm 0.009$ | $0.499 \pm 0.007$ | $0.625 \pm 0.005$ | $0.776 \pm 0.008$ | $0.547 \pm 0.006$ | $0.681 \pm 0.004$ | $0.930 \pm 0.004$ | $0.582 \pm 0.005$ |
| Ours (RFF) | $\mathbf{0.578 \pm 0.003}$ | $\mathbf{0.645 \pm 0.005}$ | $\mathbf{0.513 \pm 0.003}$ | $\mathbf{0.639 \pm 0.003}$ | $\mathbf{0.790 \pm 0.003}$ | $\mathbf{0.560 \pm 0.003}$ | $\mathbf{0.693 \pm 0.002}$ | $\mathbf{0.939 \pm 0.002}$ | $0.585 \pm 0.003$ |
| Ours (Nyström) | $0.572 \pm 0.007$ | $0.637 \pm 0.005$ | $0.508 \pm 0.008$ | $0.634 \pm 0.005$ | $0.786 \pm 0.004$ | $0.555 \pm 0.007$ | $0.685 \pm 0.006$ | $0.925 \pm 0.004$ | $\mathbf{0.587 \pm 0.007}$ |
| **Coat** Ours (SVDD) | $0.368 \pm 0.011$ | $0.382 \pm 0.012$ | $0.296 \pm 0.009$ | $0.414 \pm 0.012$ | $0.478 \pm 0.014$ | $0.332 \pm 0.009$ | $0.473 \pm 0.012$ | $0.660 \pm 0.021$ | $0.369 \pm 0.009$ |
| Ours (RFF) | $0.361 \pm 0.012$ | $0.384 \pm 0.022$ | $0.282 \pm 0.015$ | $0.387 \pm 0.005$ | $0.454 \pm 0.030$ | $0.309 \pm 0.010$ | $0.468 \pm 0.008$ | $\mathbf{0.671 \pm 0.036}$ | $0.360 \pm 0.011$ |
| Ours (Nyström) | $\mathbf{0.389 \pm 0.011}$ | $\mathbf{0.390 \pm 0.017}$ | $\mathbf{0.314 \pm 0.012}$ | $\mathbf{0.427 \pm 0.023}$ | $\mathbf{0.495 \pm 0.038}$ | $\mathbf{0.347 \pm 0.019}$ | $\mathbf{0.486 \pm 0.015}$ | $0.653 \pm 0.030$ | $\mathbf{0.384 \pm 0.013}$ |

## D  Experiments Compute Resources

We conduct all experiments on a server with 112-core Intel(R) Xeon(R) Gold 6330 CPU @ 2.00GHz. The server is equipped with a 512GB random access memory (RAM). To reproduce all the experimental results including the baselines takes a few hours.

