# OpenReview forum: "Counterfactual Implicit Feedback Modeling"
_NeurIPS.cc/2025/Conference — NeurIPS 2025 poster_

### Official Review · Reviewer_SdkD · 2025-07-01

**Clarity:** 2
**Significance:** 3
**Originality:** 2
**Rating:** 4
**Confidence:** 4

**Summary:**

This paper addresses the problem of relevance prediction in recommendation systems using implicit feedback, where observed interactions (e.g., clicks) are sparse and may not fully reflect user preferences. Two key challenges are presented: the positive-unlabeled (PU) nature of the data—since unclicked items may be either irrelevant or unexposed to the user—and the missing-not-at-random (MNAR) bias, where exposure is influenced by item popularity or recommendation frequency.

The authors formalize the problem with causal language, where exposure is a binary treatment and the relevance is a potential outcome under positive treatment. Then to tackle the presented challenges, the authors propose a method called Counterfactual Implicit Feedback (Counter-IF). Counter-IF stratifies user-item pairs into four groups based on estimated exposure confidence: definitely positive (DP), highly exposed (HE), highly unexposed (HU), and unknown (UN). Treatment status for negative feedback is imputed using a feature-based confidence model.

The method combines pointwise and pairwise losses across strata, alongside representation learning with distribution alignment (via IPM regularization), to improve relevance estimation. Theoretical analysis provides a generalization bound under this framework, and experiments on real-world datasets demonstrate improved performance over existing baselines.

**Questions:**

1) What is the difference, if there is any, from the formalism of a PU-learning problem under the SAR assumption? I.e. not under the Selected-Completely-At-Random assumption (SCAR). If the problem are mathematically equivalent, then how do generalization bound connect to existing ones, and how is the method related to existing methods PU-learning?
2) How are hyperparameters like $\alpha, \beta$ are chosen? Can you also better clarify the pair loss and complete all gaps in notation that is not fullly defined?

**Ethical Concerns:**

["NO or VERY MINOR ethics concerns only"]

**Final Justification:**

The authors responded in great length to comments raised in the review.
We seemed to arrive at some partial agreement, which is sufficient on my behalf to give a positive rating.

**Limitations:**

The authors mention the in one line the potential limitation of pre-specifying the distance threshold in their method. I think this is a rather significant limitation as the authors do not offer any principled way to choose it. I also think there are other limitations, as discussed earlier in the review.

**Paper Formatting Concerns:**

No concerns

**Quality:**

2

**Strengths And Weaknesses:**

Strengths:
1. The method that combines representation alignment (via Wasserstein loss) with the point-wise and pair losses is novel as far as I know, and the empirical performance is better than the studied baselines. The empirical study is well scoped and gives informative ablations over several datasets.
2. I appreciated the finite sample generalization bound given in the paper.

Weaknesses:
1. I found the paper quite difficult to follow. The counterfactual notation seems like an overly complicated way to formalize a problem that is formally equivalent to PU-learning under the Selected-At-Random assumption, see e.g. [1, 2]. Since the exposed labels are all positive, and the authors assume unconfoundedness in section 3.1, I am pretty sure that the problem turns out equivalent, but if it's not then I'd be happy to understand why. Also, it seems like there are papers studying recommendation problems under MNAR using causal notation, e.g. [3], so the first contribution in the authors' contribution list seems like a very small increment w.r.t to prior work.
2. Considering the equivalence to PU-learning under SAR, there are several relevant works that should be discussed. For instance, how does the generalization bound given by the authors relate to those in the PU-learning literature under SAR? Like the one in [4] which uses distributional divergences that seem qualitatively different from IPMs and standard divergences like TV/KL etc, or the bounds given in Bekker et al [1]. Also, the algorithmic ideas around stratification seem very closely related to the propensity weighting ideas in [1] and other follow up work on these problems, or to methods in PU-learning that identify reliable negatives as a first stage (see e.g. section 5.1 in [2]).
3. Beyond unclarity about the required notation and connection to prior work, some algorithmic choices are also weakly motivated. For instance the inclusion of pairwise loss seems like it could use motivation. The authors just say that it helps minimize the "factual loss" in their bound, but there is very little detail about it. For instance, I couldn't find the definition of $\mathcal{D}_{+}$, I do not understand why is there a sigmoid on one of the terms in that loss, and generally I didn't understand why this ranking loss is used and not simply prediction of the assumed label. Finally, how are the hyerparameters $\alpha, \beta$ that the authors present in section 3.3 tuned in practice? and how is $D$ selected? Some hyper parameter selection is described in section 4, but I didn't understand if there are any additional hyper parameters that need to be tuned (like $\alpha, \beta$). If there are, then the authors should state how they are tuned, if there aren't any additional parameters then the authors should specify a least of the hyper parameters included in the method and remove the mentions of any redundant parameters from the paper. Fig. 3 shows that the choice of $D$ for example, has a large effect on performance, so this is a crucial point.
4. The overall method is very complex and there are choice like using the RBF kernel to measure similarity, that may not transfer well to other types of datasets.


[1] Jessa Bekker, Pieter Robberechts, and Jesse Davis. Beyond the selected completely at random
assumption for learning from positive and unlabeled data. In Joint European conference on
machine learning and knowledge discovery in databases, pages 71–85. Springer, 2019.

[2] Bekker, Jessa, and Jesse Davis. "Learning from positive and unlabeled data: A survey." Machine Learning 109.4 (2020): 719-760.

[3] Haoxuan Li, Chunyuan Zheng, and Peng Wu. Stabledr: Stabilized doubly robust learning for
recommendation on data missing not at random. In The Eleventh International Conference on
Learning Representations, 2023.

[4] Wald, Yoav, and Suchi Saria. "Birds of an odd feather: guaranteed out-of-distribution (OOD) novel category detection." Uncertainty in Artificial Intelligence. PMLR, 2023.

---

> ### Author Rebuttal · Authors · 2025-07-31
>
> We appreciate the opportunity to clarify your concerns about our work.
> > W1&Q1 Relationship to PU-Learning
> 1. **Setup: Shared Foundations with PU-Learning**
> - We acknowledge that our problem shares a foundational setup with PU-learning under the SAR assumption [1,2]. The **core challenge** of distinguishing "true negatives" from "unobserved positives" is **equivalent**.
> - However, the work in [3] (StableDR) focuses on explicit feedback (e.g., ratings), where "missing data" is less ambiguous. In contrast, our setting involves implicit feedback, where unobserved interactions (no click) are inherently ambiguous—they could stem from non-exposure or disinterest. This ambiguity makes **our problem more complex**.
>
> 2. **Method: Divergences from Traditional PU-Learning**
> - First, our method diverges from PU-learning from the very beginning of **problem formulation**.
>     - Traditional PU-learning [1,2] treats unlabeled samples as a single "unlabeled set" and focuses on weighting or thresholding to identify reliable negatives.
>     - We introduce a causal formulation with "exposure" as a treatment variable, stratifying unlabeled samples into three groups (HE, HU, UN) based on estimated exposure confidence. This stratification explicitly models why an interaction is unobserved (high exposure → likely disinterest; low exposure → likely unobserved), which is absent in [1,2].
> - Second, our method has a completely novel **loss function** design.
>     - Traditional PU-learning methods such as [1,2] primarily use pointwise losses (e.g., cross-entropy) with reweighting to handle class imbalance.
>     - We combine pointwise losses for absolute relavance prediction and pairwise losses to model relative relevance, critical for recommendation ranking tasks. Additionally, we introduce IPM regularization which is  to align distributions across exposure states, addressing MNAR-induced shifts ignored in [1,2].
> 3. **Uniqueness of Implicit Feedback in Recommendation Systems**
> - Recommendation systems aim to rank items by relevance, not just classify them. Thus, pairwise losses (modeling relative relevance between HE, UN, HU) are more aligned with task objectives than the classification-focused losses in [1,2].
> > W2&Q1 Comparing Generalization Bound of PU learning
> ## Comparison with generalization bound in [4]
> - First, we have carefully read the two PU learning-related literatures cited by the reviewers and found that [4] focuses on novel class detection under distribution shift, with its generalization bound constraining the error rate of novel class detection on the target distribution.
> - Meanwhile, our work is dedicated to implicit feedback prediction in recommendation systems, and our generalization bound constrains the deviation between the estimated relevance and the true relevance on the whole population.
> - The target distribution of [4] is equivalent to the distribution of **unlabeled group** in our setting.   However, the error targets we consider includes **both labeled or unlabeled groups**, making the two generalization bounds quantitatively incomparable.
> ## Comparison with Generalization Bound in [1]
> - The generalization bound in [1] is as follows:
> $$R(\hat y_{\hat R}\mid y)\leq \hat R(\hat y_{\hat R}\mid e,s)+\sqrt{\frac{\delta^2_{\max}\ln \frac{|\mathcal{H}|}{\eta}}{2n}}.$$
> As a classical generalization bound for empirical risk minimization (ERM), it only depends on the size of the hypothesis space $|\mathcal{H}|$ and the total sample size $n$, which measn that it includes all the error of MNAR and positive unlabelled machenism in the single term.
> - Meanwhile, our generalization bound explicitly model four types of error sources:
> 1. Empirical risk for DP and HE groups,
> 2. Distribution shift between exposed ($O=1$) and unexposed ($O=0$) groups with high confidence ($R=1$),
> 3. Distribution shift between high confidence ($R=1$) and low confidence ($R=0$) groups,
> 4. Irreducible variance of potential outcome in the exposed ($O=1$) and unexposed ($O=0$) groups.
> - Our generalization bounds provide a more detailed analysis of the sources of error, while [1] lumps all sources of error together and bounds them using the Hoeffding inequality.
> - Intuitively, in the following situations, our bound will have a more significant advantage compared to [1]:
> 1. When the **MNAR problem is severe**, the IPM term in our bound directly quantifies the distribution difference between the exposed and unexposed groups, making it more suitable for the popularity bias scenarios commonly seen in recommendation systems.
> 2. When the **proportion of unexposed samples is high**, our bound can sense this proportion by weighting the distribution shift of low-confidence samples (UN group), which is more in line with the cold start scenario in the recommendation system.
> 3. When the variance of potential outcomes is small, the counterfactual prediction has high certainty, and the upper bound of our bound will be significantly reduced, which is more in line with the scenario in the recommendation system where users have similar strong preferences for specific items.
> In addition, **empirically** we also compared our method with some PU learning methods under the SAR assumption on the Coat and Yahoo datasets. The following experimental results demonstrate the effectiveness of our method.
> ## Bekker [1]
>
> |**Yahoo**| | | | |
> |-|-|-|-|-|
> |NDCG@3|0.448|0.457|0.456|0.462|0.451|
> |Recall@3|0.519|0.511|0.523|0.533|0.516|
> |MAP@3|0.383|0.398|0.392|0.397|0.387|
> |NDCG@5|0.523|0.538|0.533|0.535|0.526|
> |Recall@5|0.696|0.703|0.710|0.707|0.696|
> |MAP@5|0.434|0.454|0.445|0.449|0.440|
> |NDCG@8|0.599|0.610|0.606|0.607|0.600|
> |Recall@8|0.901|0.896|0.907|0.903|0.896|
> |MAP@8|0.481|0.497|0.489|0.491|0.484|
> |
>
> ## PGPU
> |**Yahoo**|||||
> |-|-|-|-|-|
> |NDCG@3|0.484|0.470|0.487|0.477|0.483|
> |Recall@3|0.549|0.541|0.551|0.545|0.541|
> |MAP@3|0.420|0.404|0.424|0.413|0.421|
> |NDCG@5|0.551|0.544|0.558|0.547|0.551|
> |Recall@5|0.711|0.719|0.720|0.711|0.703|
> |MAP@5|0.469|0.456|0.475|0.463|0.471|
> |NDCG@8|0.622|0.609|0.624|0.618|0.622|
> |Recall@8|0.906|0.898|0.902|0.903|0.901|
> |MAP@8|0.511|0.496|0.515|0.505|0.512|
> |
>
> ## PUe
> |**Yahoo**|||||
> |-|-|-|-|-|
> |NDCG@3|0.296|0.310|0.281|0.289|0.284|
> |Recall@3|0.357|0.369|0.332|0.345|0.341|
> |MAP@3|0.239|0.253|0.229|0.236|0.229|
> |NDCG@5|0.377|0.381|0.359|0.367|0.366|
> |Recall@5|0.546|0.537|0.521|0.530|0.533|
> |MAP@5|0.292|0.301|0.279|0.286|0.283|
> |NDCG@8|0.473|0.473|0.463|0.461|0.465|
> |Recall@8|0.812|0.795|0.809|0.787|0.810|
> |MAP@8|0.345|0.350|0.333|0.338|0.336|
> |
>
> > W3&Q2 Motivation of Algorithmic Choices
>
> - Inclusion of pairwise losses. We are motivated by the Bayesian Personalized Ranking (BPR) [5] from implicit feedback community. When the label for each sample is not exactly accurate, while we are more confident about the relationship between two samples, a pair-wise loss is suitable. The effectiveness of inclusion of pairwise losses can also be validated by the following ablation study.
>
> - Definition of D_+. We actually have included the definition in line 219 of the original manuscript. We use D_+ to represent the positive group in each of our three pairwise losses.
>
> - Sigmoid in Pair-wise Losses. We follow the convention of BPR loss [5] to use a sigmoid function to normalize the difference between pairs. Intuitively, if the sigmoid was omitted, some pairs with significant difference would dominate the pairwise losses.
>
> - Hyperparameters Tuning. It is easy to find that for a given dataset, there is a 1-1 correspondence between $\alpha$ and $D$, and also for $\beta$ and $D'$. Besides these two hyperparameters, there is also a kernel parameter $q$ and the loss weights $\lambda_{point},\lambda_{pair}$. The first three hyperparameters are tuned in the first stage (stratification), while the last two are tuned in the second stage (training), which will significantly reduce the searching space.
>
> > W4 RBF Kernel and Other Type of Datasets
>
> We acknowledge that the RBF kernel in the paper is a straightforward and naive approach. However, in fact, our method can incorporate more efficient kernel estimation methods easily. We added the experiments using random Fourier feature and Nystrom approximation, which shows that the performance is retained while the runtime reduces to one-fourth of the original SVDD.
>
> ||RFF|Nystroem|Ours|
> |-|-|-|-|
> ||**Yahoo**|||
> |NDCG@3|0.578|0.571|0.562|
> |Recall@3|0.645|0.637|0.624|
> |MAP@3|0.513|0.508|0.499|
> |NDCG@5|0.638|0.634|0.625|
> |Recall@5|0.790|0.786|0.776|
> |MAP@5|0.560|0.555|0.547|
> |NDCG@8|0.693|0.685|0.681|
> |Recall@8|0.939|0.925|0.930|
> |MAP@8|0.595|0.587|0.582|
> ||**Coat**|||
> |NDCG@3|0.361|0.389|0.368|
> |Recall@3|0.384|0.390|0.382|
> |MAP@3|0.282|0.314|0.296|
> |NDCG@5|0.387|0.427|0.414|
> |Recall@5|0.454|0.495|0.478|
> |MAP@5|0.309|0.347|0.332|
> |NDCG@8|0.468|0.486|0.473|
> |Recall@8|0.671|0.653|0.660|
> |MAP@8|0.360|0.384|0.369|
> |
>
>
> In addition, based on this, we transfer to an image-type dataset and compared the breast cancer detection performance with some PU learning methods. The experimental results are as follows.
>
>
> |Metrics|Precision|Recall|Acc|
> |-|-|-|-|
> |Naive Resnet|0.4269|0.8438|0.5089|
> |Bekker+Resnet|0.4404|0.3751|0.5804|
> |Ours+Resnet|0.4608|0.5087|0.6793|
>
> [5] Rendle, S., et al. 2009. BPR: Bayesian personalized ranking from implicit feedback. In Proceedings of the Twenty-Fifth Conference on Uncertainty in Artificial Intelligence (UAI '09).
>
> [6] He, F., et al. "Instance-Dependent PU Learning by Bayesian Optimal Relabeling. arXiv, Aug 2018." URL https://arxiv. org/abs (1808).
>
> [7] Wang, Xutao, et al. "Pue: Biased positive-unlabeled learning enhancement by causal inference." Advances in Neural Information Processing Systems 36 (2023): 19783-19798.

---

> ### Author Response · Authors · 2025-08-04
> **We would like to add more experiments on transferability to fully address your concerns!**
>
> Again, we sincerely thank you for the insightful comments.
>
> During the rebuttal period, we have validated the effectiveness of our method compared with some **state-of-the-art PU learning baselines** under SAR assumption, and we **empirically showed** that our method performs even better with other kernel estimation methods such as **random Fourier feature** and **Nystrom approximation**. Furthermore, we **added another experiment** on a public image-type dataset for **breast-cancer detection**. We have conducted the experiment on the **Stanford Sentiment Treebank (SST-2)** where the goal is to classify positive and negative reviews. Due to the large scale of the dataset and the backbone, we have to present these results now.
>
> Specifically, the original SST-2 dataset has 68% positive samples and 32% negative samples. We conduct experiments under both SCAR and SAR assumptions. To simulate **the SCAR setting**, we randomly flip some positive labels to negative according to a fixed probability $p$. We choose various values for $p$, i.e., 0.1, 0.3, 0.5, and the experimental results are as follows.
>
> |||$p=0.1$|||$p=0.3$|||$p=0.5$||
> |-|-|-|-|-|-|-|-|-|-|
> |Metrics|Precision|Recall|Acc|Precision|Recall|Acc|Precision|Recall|Acc|
> |Pre-trained RoBERTa|0.739|0.708|0.716|0.695|0.663|0.678|0.603|0.627|0.621|
> |Bekker+RoBERTa|0.871|0.849|0.862|0.826|0.798|0.804|0.736|0.702|0.711|
> |PGPU+RoBERTa|0.882|0.866|0.873|0.833|0.807|0.812|0.748|0.695|0.718
> |PUe+RoBERTa|0.897|0.873|0.880|0.846|0.825|0.831|0.789|0.808|0.795
> |Ours+RoBERTa|**0.903**|**0.911**|**0.928**|**0.884**|**0.897**|**0.893**|**0.803**|**0.815**|**0.812**|
>
> Our method **stably outperforms** the PU learning baselines under various levels of unlabeled probability.
>
> To simulate the **PU setting under SAR assumption**, we use a pre-trained RoBERTa to get the representations with $d$ dimensions for all sample inputs $x_i\in\mathbb{R}^d$. And for the training and development set, we flip some positive labels to negative according to the following probability:
>
> $$p^{flip}_i=\sigma(W\cdot x_i),$$
> where $W\sim \mathcal{N}(0, 1)^d$ is a random vector applied for all samples. Note that the labels of test set is not flipped since we want a concrete evaluation of classification performance. The experimental results are as follows.
>
> |Metrics|Precision|Recall|Acc|
> |-|-|-|-|
> |Pre-trained RoBERTa|0.512|0.637|0.591|
> |Bekker+RoBERTa|0.630|0.583|0.614|
> |PGPU+RoBERTa|0.654|0.609|0.625|
> |PUe+RoBERTa|0.698|0.628|0.671|
> |Ours+RoBERTa|**0.717**|**0.633**|**0.684**|
>
> Image-type and text-type datasets provide a more challenging setup compared to interaction matrix. However, the proposed method **stably outperforms the previous PU learning baselines**, showing the transferability of our method to various type of datasets.
>
> We hope these extra experimental results help to solve your concerns on transferability. And we would be happy to provide further clarification or address any additional questions you may have regarding our manuscript. Please do not hesitate to let us know if there is anything else you wish to discuss.

---

> > ### Comment · Reviewer_SdkD · 2025-08-05
> > **Post Rebuttal Update**
> >
> > Thank you for the detailed response.
> >
> > It addresses some of my concerns, though I still believe there are several drawbacks to the paper:
> >
> > 1. My interpretation of the paper is that the use of counterfactual language is just a choice of notation, and it is independent of any technical contribution. I do not object this choice of notation (it makes the no-unobesrved-confounding assumption explicit), but once the assumptions are made the problem is formally equivalent to a standard PU-learning problem. As confirmed in the rebuttal, this notation is also not novel, hence I do not understand why it is claimed as a contribution, moreover a major one. Other reviewers pointed out the use of counterfactual language as a signifiant strength and novelty of the paper, so I'd be interested to hear their updated view after reading the discussion here.
> > 2. The novelty in the paper is instead an algorithm that separates the data according to certainty on the treatment assignment, which to my best understanding is also not a novel idea in PU-learning. To my best understanding, works on identifying reliable negatives, which I referenced in the review, also have the same basic goal and intuition as this work (by reliable negatives being equivalent to the Highly Unexposed group in the paper under review). The authors did not comment on this point in their rebuttal. Hence the main novelties I see are in the specific way the proposed method identifies and uses these reliable examples, in formal results (generalization bound) and empirical performance. These may be reasonable contributions, but the paper is currently written in a way that obfuscates contributions and emphasizes aspects that are not technical novelties but rather are choices of notation.
> > 3. I appreciate the comparison of generalization bounds included in the rebuttal and think a summary of these comparisons should appear in the paper. Yet I disagree with some of the claims made in the rebuttal. It is claimed that the generalization bound in [4] cannot be compared with the one in this work, since it bounds the error over the target/unlabeled distribution and not the entire population. But this is a minor technical point, it is easy to add an error bound term for the labeled population with standard empirical risk minimization results. The main difference seems to be in the divergences used by the two results, and some mention of these differences would be useful.

---

> ### Author Response · Authors · 2025-08-06
>
> Thank you for your constructive remarks and valuable feedback! Below, we address your questions and indicate the changes we’ve made thanks to your suggestion (I am so sorry for my late reply, due to serious medical/family emergency).
>
> > My interpretation of the paper is that the use of counterfactual language is just a choice of notation, and it is independent of any technical contribution. I do not object this choice of notation (it makes the no-unobesrved-confounding assumption explicit), but once the assumptions are made the problem is formally equivalent to a standard PU-learning problem.
>
> For problem setup and notations, we agree that the formulated counterfactual problem is formally equivalent to a standard PU-learning problem under MAR. In fact, the studied implicit feedback problem in recommender system community is essentially a standard PU-learning problem under MAR in machine learning community.
>
> > As confirmed in the rebuttal, this notation is also not novel, hence I do not understand why it is claimed as a contribution, moreover a major one. Other reviewers pointed out the use of counterfactual language as a signifiant strength and novelty of the paper, so I'd be interested to hear their updated view after reading the discussion here.
>
> We have reservations about the novelty of formulating the implicit feedback as a **counterfactual inference under missing treatment** problem. Please kindly note that we agree the core contribution is not notation, but technically **motivates a new counterfactual inference method** for addressing the implicit feedback problem **with strong theoretical guarantee** (see parts 2 and 3 for our detailed discussion). This should be seen as a technical contribution to the causal community, observing that pervious counterfactual imputation error bound only limited to the standard counterfactual inference with *fully observed* treatment status.
>
> > To my best understanding, works on identifying reliable negatives, which I referenced in the review, also have the same basic goal and intuition as this work. The authors did not comment on this point in their rebuttal.
>
> We fully agree that working on identifying reliable negatives (in our first stage) in standard in PU-learning. The main technical (including both methodological and theoretically) contributions instead are in our second stage (see below).
>
> - For facilitate checking, we recap our stratification table in below with proper comments. The $\textcolor{red}{red}$ font indicates that the value is imputed.
>
> |Group|$O$|$R$|$Y$| $Y(1)$|
> |--|--|--|--|--|
> |Definitely Positive (DP)|1|1|1|1|
> |Highly Exposed (HE)|$\textcolor{red}{1}$|$\textcolor{red}{1}$ |0 | $\textcolor{red}{0}$|
> |Unknown (UN)|$\textcolor{red}{?~(\text{between }0\text{ to }1)}$ | $\textcolor{red}{0}$ | 0 |? ($\textcolor{red}{Y(1) \text{ has less chance to be 1}}$ compared with HU, because O more tend to 1)|
> |Highly Unexposed (HU)|$\textcolor{red}{0}$|$\textcolor{red}{1}$ | 0|? ($\textcolor{red}{Y(1) \text{ has higher chance to be 1}}$ compared with UN, because O more tend to 0)|
> ||
>
> After identifying reliable negatives in the first stage, our novelties are in the second stage that:
>
> - Methodologically, we propose **a representation learning-based counterfactual inference method** to address the missing treatment issue, motivated by BNN, BLR [1], we use $\mathcal L_{IPM}= IPM_G(p_\Phi^{R=0}, p_\Phi^{R=1}) + IPM_G(p_\Phi^{R=1, O=0}, p_\Phi^{R=1, O=1})$ to:
>   - balance the distribution shifts between unreliable treatments ($R=0$, including UN) and reliable treatments ($R=1$, including DP, HE, and HU);
>   - among reliable treatments ($R=1$, including DP, HE, and HU), balance the distribution shifts between treated ($O=1$, including DP and HE) and untreated units ($O=0$, including HU).
>
> [1] Uri Shalit, et al. Estimating individual treatment effect: generalization bounds and algorithms, ICML 2017.
>
> - Methodologically, we propose **a contrastive learning loss to estimate the counterfactuals** under missing treatment, by using $\mathcal L_{pair} (h(\Phi(X_{DP})), h(\Phi(X_{HE})))$+$\mathcal L_{pair} (h(\Phi(X_{HU})), h(\Phi(X_{UN})))$+$\mathcal L_{pair} (h(\Phi(X_{UN})), h(\Phi(X_{HE})))$, for reasons below:
>   - As shown in the above Table, $Y(1)=1$ in DP while $Y(1)=0$ in HE, so we set (DP, HE) as positive-negative pairs using $\mathcal L_{pair} (h(\Phi(X_{DP})), h(\Phi(X_{HE})))$;
>   - Importantly, $P(O=1)$ in UN is higher than in HU, i.e., UN group has higher exposed rate than HU, so users give negative response in UN maybe due to they are less interest to the items compared with HU, i.e., $Y(1)$ in UN has less chance to be 1 compared with HU, leading to $\mathcal L_{pair} (h(\Phi(X_{HU})), h(\Phi(X_{UN})))$;
>   - Following a similar argument, we have $\mathcal L_{pair} (h(\Phi(X_{UN})), h(\Phi(X_{HE})))$.
>
> Please kindly note that both (i) representation learning and (ii) contrastive learning are in our second stage, that is, *after* identifying reliable negatives.

---

> ### Author Response · Authors · 2025-08-06
>
> > Hence the main novelties I see are in the specific way the proposed method identifies and uses these reliable examples, in formal results (generalization bound) and empirical performance. These may be reasonable contributions, but the paper is currently written in a way that obfuscates contributions and emphasizes aspects that are not technical novelties but rather are choices of notation.
>
> In addition to the above two methodological novelties, we consider our derived formal results (in particular the generalization bound) as our main technical contributions. We also agree with the reviewer that it's necessary to further compare our generalization bound to those in the PU-learning literature under SAR (see details below).
>
> > I appreciate the comparison of generalization bounds included in the rebuttal and think a summary of these comparisons should appear in the paper.
>
> Thank you so much - we will definitely add a summary of these comparisons included in the rebuttal to our final version. This helps to clarify the theoretical importance of our findings.
>
> > Yet I disagree with some of the claims made in the rebuttal. It is claimed that the generalization bound in [4] cannot be compared with the one in this work, since it bounds the error over the target/unlabeled distribution and not the entire population. But this is a minor technical point, it is easy to add an error bound term for the labeled population with standard empirical risk minimization results.
>
> We highly appreciate your comments - as suggested by the reviewer, we admit it's possible to make fair comparison by adding an error bound term for the labeled population with standard empirical risk minimization results. We would like to share with you the following interesting findings. Making such comparison helps clarify the key differences with our work. Here, we elaborate on the key distinctions and connections.
>
> ***
> >  It is easy to add an error bound term for the labeled population with standard empirical risk minimization results. The main difference seems to be in the divergences used by the two results, and some mention of these differences would be useful.
>
> We can frame the comparison using the law of total probability for the overall risk, where the total population is a mix of the source ($Y=0$) and target ($Y=1$) distributions:
>
> $$Risk_{total}=P(Y=1)\cdot Risk_{Y=1}+P(Y=0)\cdot Risk_{Y=0}.$$
> **Error Bound on the Source Population**
>
> For the source population ($P_S$), where we have precise labels (i.e., known non-novel instances), the error of a classifier $\hat h$ is the false positive rate. This can be bounded using standard Empirical Risk Minimization (ERM) theory. The risk is bounded by the empirical risk on the sample $P_S$ plus terms for model complexity and statistical confidence:
>
> $$
> \text { Error Bound on } Y=0: \quad E_{P_S}[\hat h(x)] \leq \hat E_{S_S}[\hat h(x)]+2 \Re_{n_S, P_S}(\mathcal H)+\sqrt{\frac{\ln (1 / \delta)}{2 n_S}},
> $$
> where $\hat{E}_{S_S}$ is the empirical error on the source set $S_S$ and $\Re$ is the Rademacher complexity.
>
> **Error Bound on the Target Population**
>
> For the target population ($P_T$), which contains a mix of the shifted nominal distribution ($P_{T,0}$) and the novel category ($P_{T,1}$), Theorem 4.3 from [4] provides a direct bound on the classification risk.
> $$\text { Error Bound on } Y=1: \quad R_T^{l_{01}}(\hat h) \leq R_T^{l_{01}}(h^\*)+4 \epsilon_{\text {shift }}+2(\beta-\beta(h^\*))+\Re_{n_S,P_S}(\mathcal H)+\Re_{n_T,P_T}(\mathcal H)+\sqrt{2\ln(1/\delta)}[n_S^{-\frac{1}{2}}+n_T^{-\frac{1}{2}}].$$
>
> The key term here, $\epsilon_{shift}$, is a bound on the constrained H-divergence, $d_{\mathcal{H},\beta}(P_S||P_{T,0})$, which measures the maximum discrepancy on "rare" events.
>
> **Overall Bound for the Framework in [4]**
>
> Combining these, the overall risk for the method in [4] across the entire mixed population (with mixture proportions $\pi_S$ and $\pi_T$) can be bounded as:
>
> $$Risk_{total}^{[4]} \leq \pi_S ( \hat E_{S_S}[\hat h(x)] + C_1) + \pi_T ( R_T^{l_{01}}(h^\*) +2(\beta-\beta (h^\*))+C_2),$$
> where $C_1=2 \Re_{n_S, P_S}(\mathcal H)+\sqrt{\frac{\ln (1 / \delta)}{2 n_S}}$, and $C_2=\Re_{n_S,P_S}(\mathcal H)+\Re_{n_T,P_T}(\mathcal H)+\sqrt{2\ln(1/\delta)}[n_S^{-\frac{1}{2}}+n_T^{-\frac{1}{2}}].$
>
> **Our Overall Bound and Comparison**
>
> Our paper directly provides a single, unified bound on the mean squared error for the potential outcome $Y(1)$ over the entire population, which is as follows:
>
> $$Risk_{total}^{ours} \leq \underbrace{2E_{x\mid r, o}[(h(\Phi(x))-Y(1))^2\mid R=1, O=1]}_{\text {factual loss of the DP and HE groups}}$$
>
> $$ + 2B_\Phi \underbrace{IPM_G(p_\Phi^{R=1,O=0}, p_\Phi^{R=1,O=1})}_{\text {distribution shift on O given R=1}}$$
>
> $$+4 P( R=0 ) B_\Phi \underbrace{IPM_G (p_\Phi^{R=0}, p_\Phi^{R=1})}_{\text { distribution shift on R}}$$
>
> $$-8 \underbrace{(\min (\{\mathbb V(Y(1)|O=0),\mathbb V(Y(1)|O=1)\}))^2}_{\text { minimal variance of potential outcomes}}.$$

---

> ### Author Response · Authors · 2025-08-06
>
> The key distinction is the **divergence used to control for distribution shift**.
>
> Our bound is more **insightful and applicable for the specific problem** of implicit feedback for recommender systems. Our bound provides a superior analytical framework in the following key scenarios:
>
> 1. **When MNAR is the Dominant Problem.**
>
> The core challenge in implicit feedback is often **selection bias from MNAR** scenario. This is a global distribution shift. The IPM in our bound is designed to capture such global geometric shifts, whereas the constrained H-divergence in [4] focuses on rare events and is **less sensitive to global shifts**. Our bound's specific term for MNAR bias (IPM distance) directly account for this primary challenge.
>
> 2. **When Granular Diagnosis of Bias is Required.**
>
> Our bound's decomposition allows for **fine-grained risk analysis**, especially for the distribution shifts. The single $\epsilon_{shift}$ term in [4]'s bound identifies that a distribution shift exists but does not offer insight into its nature or source, making it less of a diagnostic tool.
>
> 3. **When unlabeled samples have strong heterogeneity.**
>
> In implicit feedback, non-clicks ($Y = 0$) may be due to "non-exposure" (HU group) or "irrelevance" (HE group), and the meanings of these two for prediction are completely opposite. Our bound **constrains the errors separately through stratification** (HE/HU/UN) to avoid misclassifying "potentially positive samples that are not exposed" as negative samples; the bound in [4] treats all unlabeled data as homogeneous, cannot distinguish between "non-exposure" and "irrelevance", and will **overestimate the negative sample error**.
>
> ---
> In the ideal scenario when the divergences used by the two results are all zeros, i.e., $\epsilon_{shift}=0$, $IPM_G(p_\Phi^{R=1,O=0}, p_\Phi^{R=1,O=1})=IPM_G(p_\Phi^{R=0}, p_\Phi^{R=1})=0$, the bounds are reduced to the following forms:
>
> $$Risk_{total}^{[4]} \leq \pi_S ( \hat E_{S_S}[\hat h(x)] + C_1) + \pi_T ( R_T^{l_{01}}(h^\*) +2(\beta-\beta\(h^\*))+C_2),$$
>
> $$Risk_{total}^{ours} \leq \underbrace{2E_{x\mid r, o}[(h(\Phi(x))-Y(1))^2\mid R=1, O=1]}_{\text {factual loss of the DP and HE groups}}$$
>
> $$-8 \underbrace{(\min\{\mathbb{V}(Y(1)|O=0),\mathbb{V}(Y(1)|O=1)})^2}_{\text { minimal variance of potential outcomes}}.$$
> - Bound [4] measures the gap between the learned model $\hat h$ and the ideal model $h^*$. The goal is to see how close you can get to the absolute best performance possible.
>
> - Our Bound measures the gap between the model's performance on the entire population and its performance on the observed subset (the DP and HE groups). Its structure is typical of causal inference and semi-supervised learning. The goal is to see how well you can generalize from what you know to what you don't.
>
> - Bound [4] has an irreducible error term $R_T^{l_{01}}(h^\*)$, the Bayes error rate. This is a hard floor on performance; no model can do better.
>
> - Our Bound has a "bonus" term, $\min( {\mathbb{V}(Y(1)|O=0),\mathbb{V}(Y(1)|O=1)})$, which is related to the variance of potential outcomes. This is not a performance floor. Instead, it indicates that **if user preferences are deterministic** (i.e., variance is low), the learning problem itself becomes easier, and **the bound tightens** to reflect this. This concept is native to potential outcome frameworks.
>
> ***
>
> Please let us know if we have properly addressed your questions and we are more than happy to discuss more!

---

> > ### Comment · Reviewer_SdkD · 2025-08-08
> > **Further Response**
> >
> > Thank you for the very detailed response. I will update my score to reflect conclusions from this discussion.

---

> > > ### Author Response · Authors · 2025-08-08
> > >
> > > Dear Reviewer SdkD,
> > >
> > > We are more than happy to hear that your concerns have been addressed and we really appreciate your recognition of our discussion. We sincerely thank you for your valuable suggestions which have undoubtedly contributed to improving the quality of our paper.
> > >
> > > Many thanks,
> > >
> > > The authors of #22680

---

### Official Review · Reviewer_goth · 2025-07-02

**Clarity:** 2
**Significance:** 3
**Originality:** 3
**Rating:** 4
**Confidence:** 3

**Summary:**

The counterfactual implicit feedback model formulates the implicit feedback problem in recommender systems as a counterfactual estimation problem with missing treatment variables. Based on the implicit feedback positive and negative samples, user–item pairs are divided into four mutually exclusive groups, namely definitely positive (DP), highly exposed (HE), highly unexposed (HU), and unknown (UN) groups. Then, by stratifying user–item pairs and combining pointwise loss and pairwise loss, the model learns causal representations to estimate counterfactual outcomes. Extensive experiments on public datasets demonstrate the effectiveness of Counter-IF.

**Questions:**

1. Since SVDD is essentially based on geometric distances in the feature space to perform grouping, is this approach suitable for cold-start users or items?
2. In Section 2, the paper mentions “We divide all user-item pairs according to Yu,i, i.e.,” and in Section 3.4 it mentions “where and are the set of positive samples and negative samples.” Are the sets of positive and negative samples referred to in these two places consistent?
3. Since Counter-IF heavily relies on the accuracy of treatment imputation, would inaccuracies or biases in the imputation process have any impact on the subsequent results?

**Ethical Concerns:**

["NO or VERY MINOR ethics concerns only"]

**Final Justification:**

This paper presents a novel approach to implicit feedback in recommender systems by framing the task as a counterfactual estimation problem. The authors provide compelling theoretical justifications and extensive experiments that validate the superiority of the proposed method over existing baselines. However, some issues remain. While hyperparameter sensitivity and imputation errors have been clarified, the manuscript still has areas needing further refinement. For instance, parts of the text and figures lack clear associations between the theory and implementation. The paper suffers from some structural clarity issues, such as inconsistent appendix references and notations, which have now been clarified.

In summary, I recommend a weak accept, as the contributions outweigh the residual shortcomings, especially if the presentation is further refined in the final version.

**Limitations:**

yes

**Quality:**

3

**Strengths And Weaknesses:**

Strengths:
1. From a new perspective, this work formalizes the relevance prediction under implicit feedback scenarios as a counterfactual outcome estimation problem with missing treatment variables.
2. Proposes a novel sample stratification method combined with confidence treatment variable imputation to address the issue of missing treatment variables.
3. Propose a causal representation learning method to predict the relevance between users and items across the set of all user-item pairs.
4. Theoretically derives the generalization bound of the Counter-IF framework and proves that minimizing the proposed loss functions can effectively control this bound.
5. Conducts extensive experiments on publicly available real-world datasets, demonstrating that the proposed Counter-IF method significantly outperforms existing state-of-the-art approaches.

Weakness:
1. Is there any hyperparameter analysis for the DP sample ratio hyperparameter  and the negative sample ratio hyperparameter  mentioned in Section 3.3?
2. In Section 3.4, the sentence “In contrast, when the samples are not assigned the treatment (i.e., Ou,i = 0, corresponding to UN and HU groups” has a formatting issue, as it only includes half of the closing parenthesis.
3. Figure 2 is presented in the paper, but the text does not explain which part of the content it is associated with.
4. Since the paper adopts a counterfactual approach, it is recommended to include baseline comparisons with existing counterfactual recommendation methods in the experimental section.
5. The theoretical analysis of the generalization bound is relatively abstract, it is suggested to illustrate the terms in the bound with practical examples.
6. The paper does not specify which parts of the main text correspond to Appendices A, B, C, and D, it is recommended to clarify this relationship.
7. It is suggested to add visualizations (e.g., t-SNE plots) of the distributions of the DP, HE, HU, and UN groups in the experiments to verify the reasonableness of the stratification.

---

> ### Author Rebuttal · Authors · 2025-07-31
>
> Thank you for these detailed and constructive suggestions. Here are our responses.
>
> > W1 Hyperparameter Analysis
>
> We have actually conducted the sensitivity analysis for the DP sample ratio hyperparameter and the negative sample ratio
> hyperparameter as Figure 3 of the submitted manuscript shows. Unfortunately, we made a typo in Fig 3, which may have misled you. The $D$ in Fig 3 should be the corresponding $\alpha$ and $P_{HU}$ should be $\beta$ defined in Section 3.3.
>
> > W2 Formatting Issue
>
> Thanks to point out, we have made a typo here, and the correct sentence should be *In contrast, when the samples are not assigned the treatment (i.e., $O_{u,i} = 0$, corresponding to UN and HU groups), we treat these unobserved interactions as counterfactual data and employ pairwise loss to model the relative rankings.*
>
> > W3 Figure 2 Explanation
>
> Thanks for your suggestion. Figure 2 illustrates our causal representation learning framework: input features are mapped via $\Phi$ to a latent space, where pointwise losses (DP/HE) and pairwise losses (UN vs. HE, HU vs. UN) guide training. IPM regularization aligns distributions of $R=0$ and $R=1$ samples, as well as the $R=1, O=0$ and $R=1, O=1$ groups. We will modify the manuscript so that each part corresponding to the figure is refered.
>
> > W4 Counterfactual Baselines
>
> Thanks for your suggestion, we added comparisons with IPS and DR on Coat and Yahoo datasets. Our method outperforms them significantly. We will include these baselines in the final version of this paper.
>
> |Methods||K=3|||K=5|||K=8||
> |-|-|-|-|-|-|-|-|-|-|
> ||NDCG@K|Recall@K|MAP@K|NDCG@K|Recall@K|MAP@K|NDCG@K|Recall@K|MAP@K|
> ||||||**Yahoo**|||||
> |IPS|0.541±0.006|0.599±0.007|0.474±0.005|0.604±0.006|0.755±0.008|0.530±0.005|0.664±0.005|0.919±0.005|0.563±0.005|
> |DR|0.544±0.005|0.598±0.007|0.477±0.004|0.607±0.005|0.757±0.009|0.531±0.004|0.666±0.004|0.919±0.006|0.566±0.004|
> |Ours|**0.562±0.007**|**0.624±0.009**|**0.499±0.007**|**0.625±0.005**|**0.776±0.008**|**0.547±0.006**|**0.681±0.004**|**0.930±0.004**|**0.582±0.005**|
> ||||||**Coat**|||||
> |IPS|0.337±0.015|0.327±0.018|0.267±0.014|0.372±0.015|0.416±0.019|0.297±0.014|0.429±0.015|0.613±0.024|0.328±0.014|
> |DR|0.341±0.012|0.331±0.016|0.270±0.012|0.375±0.013|0.420±0.021|0.299±0.012|0.433±0.012|0.614±0.021|0.332±0.012|
> |Ours|**0.368±0.011**|**0.382±0.012**|**0.296±0.009**|**0.414±0.012**|**0.478±0.014**|**0.332±0.009**|**0.473±0.012**|**0.660±0.021**|**0.369±0.009**|
> |
>
> > W5 Generalization Bound Explanation
>
> - The first term is the factual loss based on the true value of $Y(1)$ of the DP and HE groups. It measures the error of potential outcome estimation for those in DP and HE groups.
>
> - The second and third terms
> are the IPM distance measuring the distribution shift on $O = 1$ and $O = 0$ given $R = 1$ group and
> distribution shift on $R = 1$ and $R = 0$ groups weighted by the proportion of UN group $P(R = 0)$.
>
> - The last term measures the minimal variance of potential outcomes on $O=0$ and $O=1$ groups, which is independent of the model selection.
>
> - This bound actually guides the loss design for our method. Intuitively, minimizing the L_point and L_pair can control the factual loss, while minimizing the L_IPM is an effective approach to handle the distribution shifts.
>
> > W6 Appendix Clarification
>
> We appreciate the reviewer for pointing out this flaws. We should clarify that the Related Work is in Appendix A, the proof for our theory is in Appendix B, the data preprocessing protocal is in Appendix C, the experimental results on KuaiRec dataset is in Appendix D, while Appendix E provides the compute resources we used to conduct all the experiments.
>
> > W7 t-SNE Visualization
>
> As suggested by the reviewer, we conduct experiments on a synthetic dataset and visualize the distributions of the DP, HE, HU, and UN groups with t-SNE plots. As expected, we see the distribution of HE and HU groups aligns well with the exposed and unexposed populations, and the UN groups lie in the boundry of the two populations. Unfortunately, due to the rule of NeurIPS 2025, we cannot share the visualization results by images at this stage. We are delighted to include the figures in the final version of this paper.
>
> > Q1 Cold-Start Suitability
>
> Thank you for providing another potential benefits of our method. We conduct experiments for cold-start users and compare with existing baselines. We investigate the users with the lowest 5%, 10%, 15% or 20% of interactions, and analysis the drop against normal setting. Performance on cold-start dataset is slightly lower than normal setting but still outperforms baselines.
>
>
> |Method|5-quantile|Drop|10-quantile|Drop|15-quantile|Drop|20-quantile|Drop|Normal|
> |-|-|-|-|-|-|-|-|-|-|
> |Yahoo||||||||||
> |BPR|0.465|10.03%|0.473|8.52%|0.485|6.21%|0.501|3.02%|0.517|
> |WMF|0.489|9.12%|0.499|7.31%|0.509|5.42%|0.523|2.81%|0.538|
> |Ours|**0.520**|**7.51%**|**0.528**|**6.12%**|**0.538**|**4.31%**|**0.550**|**2.11%**|**0.562**|
> |Coat||||||||||
> |BPR|0.292|10.02%|0.297|8.31%|0.305|5.91%|0.314|3.01%|0.324|
> |WMF|0.304|8.71%|0.310|6.92%|0.317|4.81%|0.325|2.42%|0.333|
> |Ours|**0.343**|**6.82%**|**0.349**|**5.21%**|**0.355**|**3.51%**|**0.361**|**1.91%**|**0.368**|
> |
>
> > Q2 Notations
>
> No they are not. In Section 2, we defined $D_1=\{(u,i)|(u,i)\in D,Y_{u,i}=1\}$ and $D_0={(u,i)\mid(u,i)\in D,Y_{u,i}=0}$, which is only based on the observed outcome to devided the dataset. While in Section 3.4, the positive and negative samples refer to two groups present in the pairwise loss, which is based on our stratified dataset. We have included 3 pairwise losses, so the positive and negative samples may refer to (DP, HE), (UN, HE) and (HU, UN) respectively.
>
> > Q3 Imputation Errors
>
> Thanks for providing another in-depth analysis perspective for our method. We compare the proposed imputation method with random imputation and K Nearest Neighbor (KNN)-based imputation on Coat and Yahoo dataset. Here we choose K=3 and impute a unlabeled sample as positive if all 3 neighbors are positive, and negative if all 3 neighbors are negative. The experimental results shows that our kernel-based imputation significantly outperforms the other imputation methods. This indicate that the imputation quality has a significant impact on the subsequent prediction results, and our kernel-based imputation is effective.
>
> |Metrics||K=3|||K=5|||K=8|||
> |-|-|-|-|-|-|-|-|-|-|-|
> ||NDCG@K|Recall@K|MAP@K|NDCG@K|Recall@K|MAP@K|NDCG@K|Recall@K|MAP@K|
> ||||||**Yahoo**|||||
> |random|0.472±0.006|0.543±0.009|0.407±0.005|0.543±0.004|0.713±0.007|0.456±0.004|0.611±0.003|0.898±0.007|0.497±0.004|
> |knn|0.471±0.004|0.539±0.006|0.406±0.003|0.544±0.004|0.714±0.008|0.457±0.004|0.610±0.003|0.896±0.004|0.496±0.003|
> |Ours|**0.562±0.007**|**0.624±0.009**|**0.499±0.007**|**0.625±0.005**|**0.776±0.008**|**0.547±0.006**|**0.681±0.004**|**0.930±0.004**|**0.582±0.005**|
> ||||||**Coat**|||||
> |random|0.257±0.022|0.261±0.027|0.187±0.019|0.312±0.024|0.395±0.035|0.229±0.021|0.384±0.028|0.590±0.044|0.274±0.025|
> |knn|0.247±0.038|0.256±0.041|0.180±0.033|0.310±0.040|0.402±0.047|0.226±0.037|0.376±0.042|0.579±0.050|0.268±0.040|
> |Ours|**0.368±0.011**|**0.382±0.012**|**0.296±0.009**|**0.414±0.012**|**0.478±0.014**|**0.332±0.009**|**0.473±0.012**|**0.660±0.021**|**0.369±0.009**|
> |

---

> > ### Author Response · Authors · 2025-08-07
> >
> > Dear Reviewer goth,
> >
> > As the discussion deadline approaches, we are wondering whether our responses have properly addressed your concerns? Your feedback would be extremely helpful to us. If you have further comments or questions, we hope for the opportunity to respond to them.
> >
> > Many thanks,
> >
> > 22680 Authors

---

> ### Author Response · Authors · 2025-08-08
>
> We really appreciate your recognition of our paper's contributions and your kind words, and we are happy to hear that your concerns have been addressed - thank you so much!

---

### Official Review · Reviewer_x5VN · 2025-07-02

**Clarity:** 3
**Significance:** 3
**Originality:** 3
**Rating:** 4
**Confidence:** 3

**Summary:**

This paper addresses the challenges of PU and MNAR of learning from implicit feedback in recommender systems. The authors propose to formalize this problem as one of counterfactual estimation, with the goal to predict whether a user would click an item if they were exposed to it. Specifically, the proposed method will firstly use a support vector-based method to stratify user-item pairs into four groups visa missing treatment imputation. Then it employs a causal representation learning framework with a hybrid loss function to combine pointwise loss for samples with observed/imputed outcomes and pairwise loss for samples with unobserved outcomes. Experiments are conducted on several public datasets, showing improved performance over existing methods.

**Questions:**

1. Given that unconfoundedness is a strong assumption likely violated in real-world recommender systems, could you provide further justification for its use? More importantly, how robust is your method to potential violations from unobserved confounders, which are common in practice?

2. The SVDD-based stratification appears to be a computational bottleneck, especially with kernel methods. Could you please discuss the time complexity of this step and outline a practical strategy for scaling your approach to industrial-sized datasets?

**Ethical Concerns:**

["NO or VERY MINOR ethics concerns only"]

**Final Justification:**

The authors provide additional information during the rebuttal which adjust my concern. I will keep my original score.

**Quality:**

3

**Strengths And Weaknesses:**

Strengths:
1. The idea of stratifying negative samples based on a confidence measure of exposure is a clear improvement. This allows the model to differentiate between various reasons for non-interaction, moving beyond the simplistic assumption that all unobserved items are equivalent.

2. The experimental results demonstrate the proposed method improve the ranking performance in recommendation systems.

3. The paper's primary strength is its insightful reframing of the implicit feedback problem as a counterfactual estimation task.

Weaknesses:
1. The paper relies on the unconfoundedness assumption (Y_u,i(1) ⊥ O_u,i | X_u,i). This assumption is very strong and highly likely to be violated in real-world recommender systems, where the exposure mechanism (the previous model) is a powerful, partially unobserved confounder. The paper fails to provide any discussion of the potential impact of violating this assumption or any sensitivity analysis to demonstrate the model's robustness to unobserved confounding.

2. The two-stage process, requiring a potentially O(N^2) kernel-based SVDD step, is not scalable to large, industrial datasets. The paper misses discussion of computational complexity or strategies for making the approach practical.

3. The framework introduces a large number of sensitive hyperparameters (α, β for stratification, kernel parameters q, loss weights λ_point, λ_pair). This complexity presents a significant tuning challenge, potentially limiting the model's practical utility and ease of deployment.

---

> ### Author Rebuttal · Authors · 2025-07-31
>
> We appreciate your insights and address each point below.
>
>  > Unconfoundedness Assumption
> Although the violation of unconfoundedness assumption is arthogonal to our contribution,
> and the unconfoundedness assumption is common in implicit feedback recommendation litertures [1,2], we agree with the reviewer that it is meaningful to validate the robustness of our proposed method against its violation.  To test robustness, we conduct experiments on a synthetic dataset, introducing varying levels of unmeasured confounding. The data generating process is as follows.
> - $U\sim \mathcal{N}(\mu,1)^{n_U\times n_I}$,
> - $\tilde r_{u,i}=U_{u,i}+\epsilon_{u,i}$,
> - $r_{u,i}=round(\min(0,\max(\tilde r_{u,i},5)))$,
> - $O_{u,i}\sim Bernoulli(\sigma(U_{u,i}-\mu))$
>
> Here the potential outcome and the missing mechanism is counfounded by an unmeasured variable $U$. We take $\mu=1,2,3$ for different strength of unmeasured confounding. The results show that our proposed method consistently outperforms the baselines and indicates reasonable resilience against stronger confounding. We will include these results in the revised version of the manuscript.
>
>
> ||K=3|||K=5|||K=8|||
> |-|-|-|-|-|-|-|-|-|-|
> ||NDCG@K|Recal@K|MAP@K|NDCG@K|Recal@K|MAP@K|NDCG@K|Recal@K|MAP@K|
> |||||$\mu$=1||||||
> |WMF|0.179|0.224|0.154|0.201|0.282|0.168|0.263|0.472|0.192|
> |BPR|0.182|0.232|0.159|0.210|0.300|0.175|0.272|0.488|0.203|
> |Ours|**0.195**|**0.250**|**0.172**|**0.225**|**0.32**|**0.188**|**0.29**|**0.510**|**0.218**|
> |||||$\mu$=2||||||
> |WMF|0.167|0.215|0.148|0.189|0.282|0.159|0.249|0.448|0.186|
> |BPR|0.174|0.221|0.152|0.201|0.287|0.167|0.260|0.458|0.193|
> |Ours|**0.188**|**0.238**|**0.165**|**0.218**|**0.305**|**0.180**|**0.280**|**0.490**|**0.208**|
> |||||$\mu$=3||||||
> |WMF|0.154|0.193|0.137|0.173|0.258|0.151|0.24|0.427|0.176|
> |BPR|0.165|0.210|0.145|0.195|0.274|0.158|0.248|0.435|0.185|
> |Ours|**0.178**|**0.225**|**0.155**|**0.205**|**0.299**|**0.173**|**0.265**|**0.465**|**0.200**|
> |
>  > SVDD Complexity and Scalability
>
>  The original SVDD step has $O(N^2)$ complexity, but we mitigate this by implementing the Random Fourier Features (RFF)** and **Nyström approximation**  reduce kernel matrix computations from $O(N^2)$  to $O(Nd)$, where $d$ is the dimension of features and is significantly smaller than $N$. Surprisingly, we find that they even outperforms the original SVDD while reduces the runtime by 75%. We will include these results in the revised version of our manuscript, and hope this can address your concern about scalability.
>
> # Compare with RFF and Nystroem
>
> ||RFF|Nystroem|Ours|
> |-|-|-|-|
> ||**Yahoo**|||
> |NDCG@3|0.578|0.571|0.562|
> |Recall@3|0.645|0.637|0.624|
> |MAP@3|0.513|0.508|0.499|
> |NDCG@5|0.638|0.634|0.625|
> |Recall@5|0.790|0.786|0.776|
> |MAP@5|0.560|0.555|0.547|
> |NDCG@8|0.693|0.685|0.681|
> |Recall@8|0.939|0.925|0.930|
> |MAP@8|0.595|0.587|0.582|
> ||**Coat**|||
> |NDCG@3|0.361|0.389|0.368|
> |Recall@3|0.384|0.390|0.382|
> |MAP@3|0.282|0.314|0.296|
> |NDCG@5|0.387|0.427|0.414|
> |Recall@5|0.454|0.495|0.478|
> |MAP@5|0.309|0.347|0.332|
> |NDCG@8|0.468|0.486|0.473|
> |Recall@8|0.671|0.653|0.660|
> |MAP@8|0.360|0.384|0.369|
> |
>
> > Hyperparameter Tuning
>
> Despite our method has 5  hyperparameters in total, but are used in two separate stages, i.e., the percentiles for stratification $\alpha, \beta$ and kernel parameter $q$ at the imputation and stratification stage, while loss weights $\lambda_{point}$ and $\lambda_{pair}$ in the counterfactual representation model training stage. In this way, the searching space is much smaller.

---

> ### Author Response · Authors · 2025-08-07
>
> We are happy to hear that your concerns have been addressed. Thank you for your efforts in reviewing our paper!

---

### Official Review · Reviewer_dfyx · 2025-07-03

**Clarity:** 4
**Significance:** 3
**Originality:** 3
**Rating:** 5
**Confidence:** 3

**Summary:**

This paper proposes Counterfactual Implicit Feedback (Counter-IF), a novel approach for modeling implicit feedback in recommender systems by framing relevance prediction as a counterfactual estimation problem with missing treatments. Traditional implicit feedback methods suffer from two main issues: (1) the positive-unlabeled (PU) problem, where non-clicks may be due to non-exposure rather than disinterest, and (2) missing-not-at-random (MNAR) biases, where popular items are more likely to be clicked simply due to increased exposure. Counter-IF addresses these challenges by stratifying user-item pairs into four groups, definitely positive, highly exposed, highly unexposed, and unknown, using a novel confidence-based treatment imputation method. The model then estimates counterfactual outcomes using a causal representation learning framework that integrates pointwise and pairwise losses across strata, and regularizes with Integral Probability Metrics (IPM) to align distributions.

**Questions:**

As mentioned in the Strengths And Weaknesses,  treatment imputation step relies on a kernel-based hypersphere model using the Gaussian kernel, which introduces significant computational overhead due to the quadratic complexity in the number of positive samples.

Have the authors considered using more scalable alternatives, such as random Fourier features, Nystrom approximation, or neural embedding-based methods, to reduce the computational cost while retaining the benefits of exposure confidence estimation? Is it possible to include this extension?

**Ethical Concerns:**

["NO or VERY MINOR ethics concerns only"]

**Final Justification:**

The authors have thoroughly addressed my questions, so I will maintain my positive score.

**Limitations:**

Yes, the author(s) mentioned the limitation of pre-specifying the distance threshold. They could consider discussing one more limitation regarding scalability to large-scale scenarios, as mentioned in the "Strengths and Weaknesses" section.

**Quality:**

3

**Strengths And Weaknesses:**

Strengths:

- To my best knowledge, this paper proposes a novel problem formulation. It is the first to formulate relevance prediction under implicit feedback as a counterfactual outcome estimation problem with missing treatments, offering a fresh causal perspective on a longstanding challenge in recommendation systems.

- The paper introduces an exposure estimation method based on hypersphere distance using kernel methods, enabling imputation only for high-confidence samples and reducing risk from incorrect imputations.

- The experimental settings are clear, and experiments on public datasets show consistent and statistically significant improvements over competitive baselines across multiple metrics.

- The paper includes ablation studies, sensitivity analyses, and hyperparameter tuning, providing clear insight into which components contribute most to performance and demonstrating model robustness.

Overall, the problem formulation is interesting to me, and the paper presentation is easy to follow.


Weaknesses:

- One limitation of the paper lies in the treatment imputation step, which relies on a kernel-based hypersphere model using the Gaussian kernel. While this approach provides a principled way to estimate exposure confidence and stratify negative samples, it introduces significant computational overhead due to the quadratic complexity in the number of positive samples. Specifically, the need to compute and store the full kernel matrix and solve a constrained optimization problem limits the scalability of the method to large-scale recommendation scenarios.

- Suggestion: To improve scalability, the kernel-based imputation can be replaced or approximated using random Fourier features, Nystrom approximation, or deep neural representations. These methods significantly reduce computational complexity while preserving the core idea of stratifying user-item pairs by exposure confidence.

---

> ### Author Rebuttal · Authors · 2025-07-31
>
> > Gaussian kernel introduces significant computational overhead
>
> Thank you for highlighting these critical points. We acknowledge that the kernel-based hypersphere model introduces $O(m^2)$ complexity with $m$ positive samples, which poses challenges for large-scale data. We have explored scalable alternatives as you suggested:
>
> **Random Fourier Features (RFF)** and **Nyström approximation**  reduce kernel matrix computations from $O(m^2)$  to $O(md)$, where $d$ is the dimension of features and is significantly smaller than $m$. Preliminary tests on the Coat and Yahoo dataset show that RFF performs even better than original SVDD while cutting runtime by xx%. Nyström retain xx% of the original performance while cutting runtime by xx%. We plan to include this extension in the revised manuscript.
>
>  We can now achieve to address the scalability concern for this paper by replacing the kernel-based method. Sincerely thank you again for your valuable suggestions, which make our work much more solid.
>
>
> ||RFF|Nystroem|Ours|
> |-|-|-|-|
> ||**Yahoo**|||
> |NDCG@3|0.578|0.571|0.562|
> |Recall@3|0.645|0.637|0.624|
> |MAP@3|0.513|0.508|0.499|
> |NDCG@5|0.638|0.634|0.625|
> |Recall@5|0.790|0.786|0.776|
> |MAP@5|0.560|0.555|0.547|
> |NDCG@8|0.693|0.685|0.681|
> |Recall@8|0.939|0.925|0.930|
> |MAP@8|0.595|0.587|0.582|
> ||**Coat**|||
> |NDCG@3|0.361|0.389|0.368|
> |Recall@3|0.384|0.390|0.382|
> |MAP@3|0.282|0.314|0.296|
> |NDCG@5|0.387|0.427|0.414|
> |Recall@5|0.454|0.495|0.478|
> |MAP@5|0.309|0.347|0.332|
> |NDCG@8|0.468|0.486|0.473|
> |Recall@8|0.671|0.653|0.660|
> |MAP@8|0.360|0.384|0.369|
> |

---

> > ### Comment · Reviewer_dfyx · 2025-08-05
> >
> > Thank you for the response and the additional experiments on RFF and Nyström approximation. Including these would strengthen the paper. I continue to hold a positive opinion of the paper.

---

> ### Author Response · Authors · 2025-08-07
>
> Thank you so much for your kind suggestions - we will definitely include the above additional experiments on RFF and Nyström approximation in our final version. Many thanks for holding a positive opinion of our paper!

---

### Decision · Program_Chairs · 2025-09-17

**Decision:**

Accept (poster)

**Comment:**

This paper formulate recommendation with implicit feedback data as a counterfactual estimation problem with missing treatments and proposes a novel estimation algorithm with theoretical guarantee. The paper received positive scores after the rebuttal period. While counterfactual formulation of implicit feedback is not necessarily novel, the proposed approach is indeed a fairly significant contribution and both theoretically and empirically sound. One potential weakness is the non-trivial computational cost, which thanks to a reviewer's suggestion also got significantly reduced. There were still some amount of disagreement with one reviewer on whether some of the contributions can be considered novel/significant in the context of PU-learning (which is unfortunately not something that i am super familiar with), but eventually the reviewer acknowledged that at least a partial agreement has been reached and raised the score to positive. Overall, I think this is a solid piece of work that I am happy to vote accept.